# Lattice oxygen-mediated electron tuning promotes electrochemical hydrogenation of acetonitrile on copper catalysts

Cong Wei[1,4], Yanyan Fang[1,4], Bo Liu[1], Chongyang Tang[1], Bin Dong[2], Xuanwei Yin[1], Zenan Bian[1], Zhandong Wang [2], Jun Liu[3], Yitai Qian[1] & Gongming Wang [1] ✉

Copper is well-known to be selective to primary amines via electrocatalytic nitriles hydrogenation. However, the correlation between the local fine structure and catalytic selectivity is still illusive. Herein, we find that residual lattice oxygen in oxide-derived Cu nanowires (OD-Cu NWs) plays vital roles in boosting the acetonitrile electroreduction efficiency. Especially at high current densities of more than 1.0 A cm$^{-2}$, OD-Cu NWs exhibit relatively high Faradic efficiency. Meanwhile, a series of advanced in situ characterizations and theoretical calculations uncover that oxygen residues, in the form of Cu$_4$-O configuration, act as electron acceptors to confine the free electron flow on the Cu surface, consequently improving the kinetics of nitriles hydrogenation catalysis. This work could provide new opportunities to further improve the hydrogenation performance of nitriles and beyond, by employing lattice oxygen-mediated electron tuning engineering.

Amines, as multipurpose basic chemicals, have been widely used as a critical feed for the industrial synthesis of pharmaceuticals and fine chemicals[1–6]. Over the past century, the commercial production of amines has continuously grown, exceeding 6 million metric tons per year now. Moreover, the amine market is predicted to increase annually by ~4.8% with the ever-growing global demand in the next decade[3,7,8]. Currently, the hydrogenation of nitriles is the most commonly used method in the industrial production of amines, which are typically operated at high temperatures and high-pressure conditions with flammable and explosive H$_2$ gas as the hydrogenation source[9–12]. Moreover, the hydrogenated product is usually a mixture of multiple amines, including primary, secondary and tertiary amines, which inevitably require purification treatment and consequently increase the manufacturing cost of the target amines[1,10,11,13]. In this regard, developing efficient hydrogenation methods under ambient conditions with low energy consumption and favorable selectivity is still in high demand for the amine industry.

Recently, the electrocatalytic hydrogenation of nitriles to amines with water as the hydrogen source and under room temperature working conditions has raised great interest, because it avoids the use of flammable and explosive hydrogen gas and is potentially powered by renewable electricity to further decrease the cost of amines[14–17]. As the key components in the electrocatalytic process, catalysts essentially determine the amine production efficiency and selectivity. Currently, copper-based catalysts are found to be active for the electrocatalytic hydrogenation of acetonitrile (AN) to ethylamine (EA) in alkaline or neutral conditions via a multielectron reduction process[16,17]. For example, Xia et al. found that Cu nanoparticles exhibits 94.6% EA faradaic efficiency (FE$_{EA}$) in a 1.0 M KOH solution with 8 wt% acetonitrile with a total current density of 50 mA cm$^{-2}$, while 84.6% of FE$_{EA}$ is achieved at a larger EA partial current density (-0.85 A cm$^{-2}$) with 12 wt% acetonitrile[16]. In addition, Zhang et al. obtained 94% FE$_{EA}$ in a CO$_2$ saturated 1.0 M KHCO$_3$ solution containing 0.5 M AN at ~−0.7 V versus the reversible hydrogen electrode (RHE) using a Cu catalyst[17]. Even so, their energy efficiency and faradaic efficiency are still not sufficient at high current densities. Moreover, the underlying catalytic mechanism is still illusive, especially regarding the correlation between the local fine structure and catalytic selectivity. Since the complexity of

[1]Department of Chemistry, University of Science and Technology of China, Hefei 230026, China. [2]National Synchrotron Radiation Laboratory, University of Science and Technology of China, Hefei 230029, China. [3]Institute of Solid State Physics, Hefei Institutes of Physical Science, Chinese Academy of Sciences, Hefei 230031, China. [4]These authors contributed equally: Cong Wei, Yanyan Fang. ✉e-mail: wanggm@ustc.edu.cn

hydrogenation of AN involves the processes of nitrile C≡N bond adsorption and activation, proton-coupling electron transfer, and amino C−N bond desorption, manipulating the local coordination and electronic structures of catalysts is crucial to achieve better catalytic performance. Typically, most metal catalysts, including Pt, Ni, Cu, and Pd, have superior adsorption properties toward the C≡N bonds of AN[15–18]. However, the potential-determining step of the catalytic process is hindered by the strong interaction between the adsorbed amino C−N bonds (EA*) and the metal sites, which consequently prevents the desorption of EA product[17]. Aiming at the catalytic surface of Cu, the Cu metal surface is basically flooded with abundant free electrons due to its metallic bonding characteristics, which enables strong adsorption toward amines by effective electron coupling (Fig. 1a)[19]. In this regard, it can be predicted that the electron transfer between the copper surface and the amines can be tuned by manipulating the electron confining capability of copper atoms, which in turn weakens the amine binding strength. Therefore, a foreseeable strategy is to create a negative charge center in the Cu lattices, in which the localized negative charge site can confine the surface free electron flow and thus effectively weaken the adsorption behavior of amines (Fig. 1b). Considering the strong electron pulling feature of oxygen, introducing oxygen atoms into Cu might be an effective strategy. Bearing this in mind, we use an oxide-derived copper (OD-Cu) catalyst with oxygen residual left in the copper lattice to demonstrate proof of concept.

Herein, oxide-derived Cu nanowire catalysts (OD-Cu NWs) with oxygen residuals were synthesized by a simple electrochemical reduction strategy and used for acetonitrile electrochemical hydrogenation. X-ray photoelectron spectroscopy (XPS), X-ray absorption fine structure (XAFS) spectroscopy, and ab initio molecular dynamics (AIMD) simulations clearly reveal the presence of oxygen residual in OD-Cu NWS after electrochemical reduction, which is stabilized in the form of $Cu_4$-O coordination. The prepared OD-Cu NWs displayed a maximum $FE_{EA}$ of ~97.8% at −0.32 V (vs RHE). Even at a large current density of more than 1.0 A cm$^{-2}$ at ~−0.44 V (vs RHE), 91% $FE_{EA}$ can still be maintained, suggesting its superior catalytic activity and selectivity. Meanwhile, a series of advanced in situ characterization methods, including in situ Raman spectroscopy, in situ synchrotron radiation Fourier transform infrared (SR-FTIR), and in situ synchrotron vacuum ultraviolet radiation photoionization mass spectrometry (SVUV-PIMS), were performed to reveal that the presence of residual oxygen in OD-Cu NWs effectively weakens the adsorption of EA on the surface and consequently leads to selective formation of the EA product. Furthermore, the weakened adsorption of EA and improved catalytic activity are attributed to the confined electron effect induced by lattice

oxygen residual via density functional theory (DFT) calculations. The concept of lattice oxygen-mediated electron tuning could provide valuable insights for the design of electrocatalysts for small organic molecule catalysis.

## Results

### Catalyst synthesis and morphological characterizations

The OD-Cu NWs grown on copper foam were prepared by electrochemical reduction methods, as illustrated in Fig. 2a. $Cu(OH)_2$ nanowire precursors were first synthesized by chemical oxidation of Cu foam, which was followed by thermal annealing in argon to obtain CuO nanowires[20,21]. Then, two approaches were employed to reduce the prepared CuO nanowires. To obtain metallic copper with limited oxygen residual, in situ electrochemical reduction was used to prepare metallic copper with abundant oxygen residuals (denoted as OD-Cu NWs) based on cyclic voltammograms measured in KOH electrolyte for CuO (Supplementary Fig. 1), while metallic copper with limited oxygen residual (denoted as Cu NWs) was achieved by thermal reduction in 3% $H_2$/Ar at 350 °C based on the $H_2$-temperature programmed reduction ($H_2$-TPR) tests (Supplementary Fig. 2). Scanning electron microscopy (SEM), transmission electron microscopy (TEM) and X-ray diffraction (XRD) were performed to acquire morphological and structural information on the synthesis process (Supplementary Figs. 3–5). In comparison with the CuO nanowire precursor, both of the prepared OD-Cu NWs and Cu NWs can maintain their initial nanowire morphology with an average diameter of ~200 nm (Fig. 2b, c and Supplementary Fig. 5).

To reveal the crystalline feature of OD-Cu NWs and Cu NWs, high-resolution transmission electron microscopy (HRTEM) was performed (Fig. 2d, e). Both Cu NWs and OD-Cu NWs display typical face-centered cubic (fcc) structures. Interestingly, the (111) facet interplanar spacing distance is increased in OD-Cu NWs. Meanwhile, the XRD analysis in Fig. 2f reveals that the diffraction peaks of OD-Cu NWs are slightly shifted to a lower angular region (0.2090 nm vs 0.2087 nm based on Bragg equation), also suggesting lattice expansion in OD-Cu NWs. Furthermore, aberration-corrected high-angle annular dark-field scanning transmission electron microscopy (HAADF-STEM) and geometric phase analysis (GPA) are further performed to study the lattice expansion[22,23]. As shown in Fig. 2g, h, the average interplanar spacings of the (111) facets of OD-Cu NWs are expanded, which is consistent with the XRD and HRTEM results. In addition, GPA analysis shows more positive strain tensor values (corresponding to the $e_{xx}$ direction) in OD-Cu (111) NWs than in Cu (111) NWs, demonstrating the increased surface tensile stress.

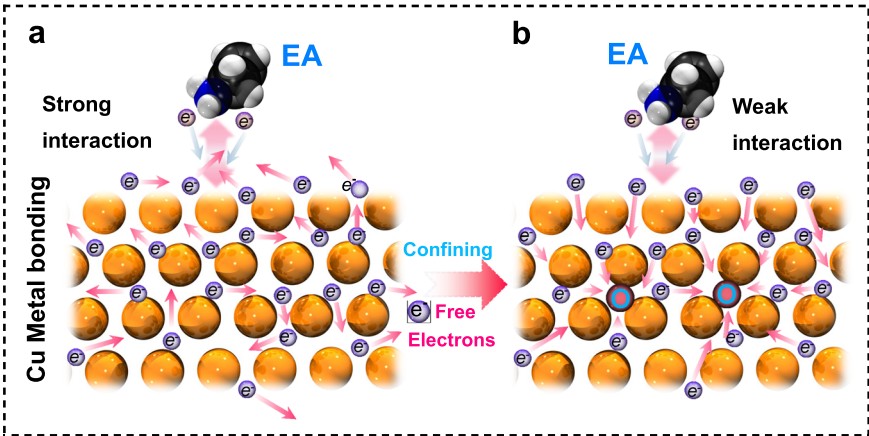

**Fig. 1 | Schematic diagram of the free electron regulation for the metallic Cu surface. a** Cu metal catalysts with a surface flooded with a large number of free electrons exhibit a strong interaction between EA and metallic Cu. (Orange, blue, black, and white spheres: Cu, N, C, and H atoms). **b** Cu catalysts with oxygen residual that confine the flow of free electrons on the Cu surface to achieve a weak interaction between EA and Cu.

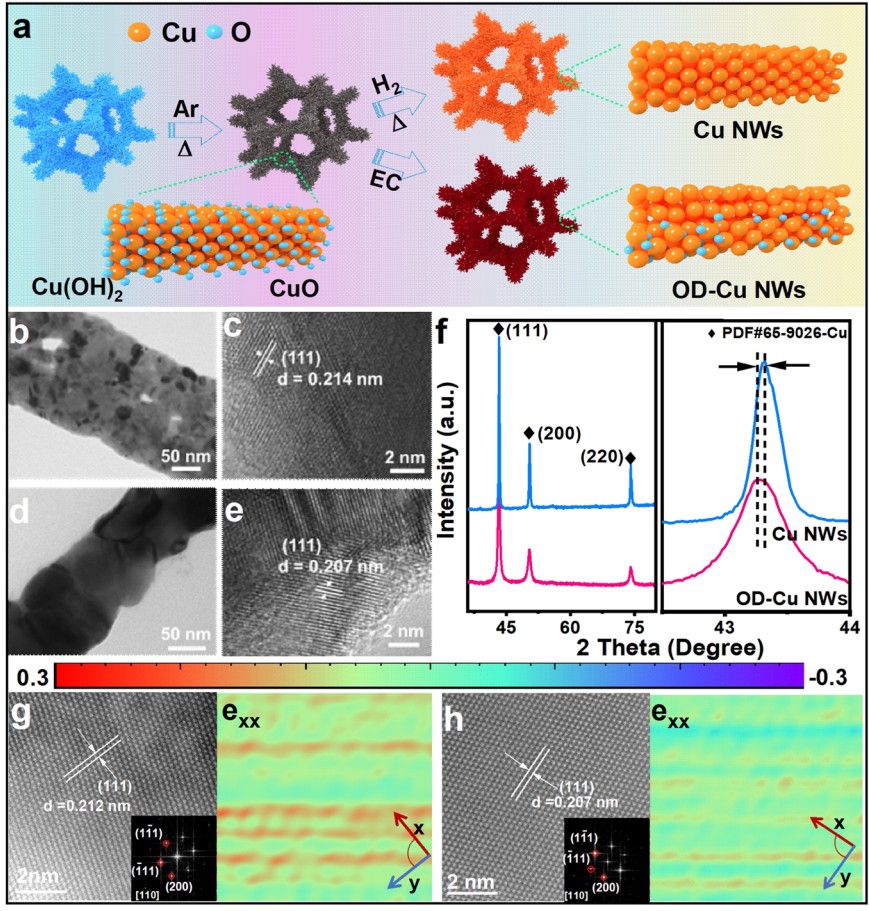

**Fig. 2 | Morphological and structural characterization. a** Schematic illustration of the synthesis of Cu NWs and OD-Cu NWs. **b**–**e** TEM and HRTEM images of OD-Cu NWs (**b**, **c**) and Cu (**d**, **e**), respectively. **f** XRD patterns of Cu NWs and OD-Cu NWs. **g**, **h** The HADDF-HRTEM images and the corresponding strain tensor mapping of OD-Cu NWs (**g**) and Cu NWs (**h**).

To further probe the origin of lattice expansion and the chemical states of the OD-Cu NWs, Raman spectroscopy, X-ray photoelectron spectroscopy (XPS), and X-ray absorption fine structure (XAFS) spectroscopy were conducted. Figure 3a shows the Raman spectra of Cu NWs and OD-Cu NWs. Typically, metallic Cu NWs do not have Raman bands, since it is Raman inactive. Notably, a weak band located at approximately 389 cm$^{-1}$ was observed in OD-Cu NWs, corresponding to a multi-phonon process stemming from the lattice oxygen in copper, demonstrating the existence of residual lattice oxygen in OD-Cu NWs[24–26]. In addition, the core-level XPS O 1 s spectra of Cu NWs and OD-Cu NWs are shown in Supplementary Fig. 6. Basically, the peak at 530.6 eV is attributed to the lattice metal-O bond, while the peaks at 531.9 and 533.3 eV are assigned to the surface adsorbed-oxygen species, including surface metal-O bonds and other surface adsorbents (such as adsorbed water), respectively[27–29]. For Cu NWs, only surface oxygen is observed, which arises from surface adsorption, while OD-Cu NWs show obvious lattice metal-O bonds, demonstrating the presence of unreduced lattice oxygen residuals. To further probe the chemical environment of O inside the lattice, coordination-sensitive O K-edge near-edge X-ray absorption fine structure (NEXAFS) spectroscopy was operated as shown in Fig. 3b. For better comparison, the standard copper oxides (CuO, and Cu$_2$O) and Cu NWs are used as references. It is worth noting that OD-Cu NWs possess a characteristic peak at 533.28 eV, due to the O 1 s to O 2$p$–Cu 3$d$ transition, which is a similar O-Cu coordination configuration as Cu$_2$O (as shown in the inset of Fig. 3b), different from the oxygen signals in CuO and Cu NWs[30–34]. However, compared with Cu$_2$O, the O 2$p$–Cu 3$d$ unoccupied state of OD-Cu

NWs is slightly upshifted by 0.25 eV, indicating that the overall crystal feature of OD-Cu NWs is different from the oxide phase.

Since it is difficult for XPS to essentially distinguish the oxidation state of Cu$^0$ or Cu$^+$, Cu L$_3$M$_{45}$M$_{45}$ Auger spectra were performed to overcome this limitation (Supplementary Fig. 7). As shown in Supplementary Fig. 8, the peak at 916.5 eV is the characteristic peak of Cu$^{\delta+}$, which is the dominant species in OD-Cu NWs, while for Cu NWs, the Cu$^0$ located at 918 eV becomes significant[25,28,35–37]. Considering that catalytic reactions basically occur at the several outermost atomic layers, precisely uncovering the surface chemical state is of high importance. Synchrotron radiation photoelectron spectroscopy (SRPES) with adjustable photon energy could be a superior technique to probe the depth-dependent electronic state evolution. Figure 3c illustrates the Cu 3 p SRPES spectra with different photon energies. The doublet peaks located at ~74.9 eV and ~77.2 eV can be attributed to Cu 3$p^{1/2}$ and 3$p^{3/2}$ of metallic Cu[38,39]. Impressively, the binding energies of Cu 3$p$ show a positive shift of approximately 0.25 eV and remain constant at 780 eV with the increase in photon energy from 180 to 380 eV, indicating the existence of Cu atoms with higher valence inside the lattice of OD-Cu NWs.

To gain insight into the local coordination environment of the Cu-based materials, X-ray absorption fine structure (XAFS) spectroscopy combined with AIMD simulations are further conducted. Figure 3d shows the Cu K-edge X-ray absorption near edge spectroscopy (XANES) spectra of OD-Cu NWs and Cu NWs as well as Cu$_2$O, CuO, and Cu foil used as references. The absorption edge at ~8978 eV originates from the 1 s–4 p transition, as enlarged in Fig. 3d, inset, where the absorption-edge position of the Cu NWs almost overlaps with the Cu

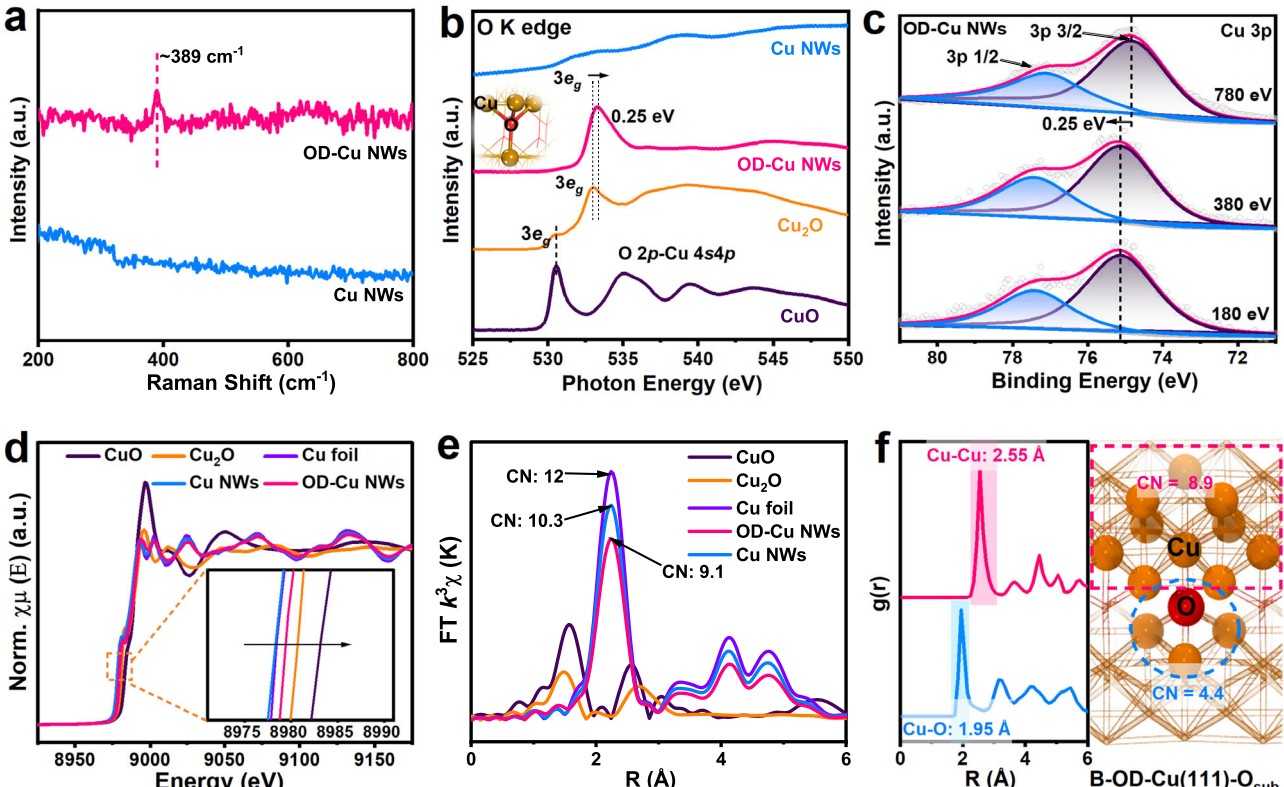

**Fig. 3 | The chemical and electronic state investigation. a** Raman spectra of OD-Cu NWs and Cu NWs. **b** O K-edge NEXAFS of Cu NWs, OD-Cu NWs, Cu₂O and CuO. **c** Cu 3p SRPES with different photon energies. **d**, **e** The Cu K-edge XANES spectra (**d**) and the corresponding FT-EXAFS spectra (**e**). Inset, enlarged view of the pre-edge energy range of samples. **f** AIMD simulation of RDFs between Cu-Cu and Cu-O on the B-OD-Cu(111)-O$_{sub}$ surface and the corresponding local coordination configurations.

foil, because of its metallic feature[40]. However, the adsorption-edge of OD-Cu NWs is blueshifted, indicating increased valence state and decreased 4 p electron density due to the presence of residual oxygen as the electron acceptor. Meanwhile, it is interesting to find that the XANES spectrum of OD-Cu NWs is located between that of Cu NWs and Cu₂O with a valence of 0.46, indicating a mixed Cu local environment (Supplementary Fig. 9). To more precisely reveal the local structure, linear composition fitting (LCF) of the catalyst was conducted as displayed in Supplementary Table 1 and Supplementary Figs. 10, 11. The Cu NWs is well fitted to the 100% Cu foil standard, while the spectrum of OD-Cu NWs is well fitted to 27.8% Cu₂O with Cu₄-O coordination and 72.2% Cu foil, indicating the existence of Cu₄-O coordination in OD-Cu NWs, which is in accordance with the O-K edge NEXAFS analysis. However, the oxygen content (12.2 at%) is too low to display significant Cu-O bonding difference in its R space. Furthermore, the local coordination of Cu is analyzed by the Fourier-transformed (FT) k³-weighted Cu K edge extended X-ray absorption fine structure (EXAFS) (Supplementary Fig. 12). As shown in Fig. 3e, the Cu-Cu coordination shell of OD-Cu NWs typically located at ~2.23 Å is significantly lower than those of Cu foil and Cu NWs, and no Cu-O bond exists, suggesting increased Cu-Cu coordination unsaturation in the presence of trace oxygen residues. To accurately obtain the average coordination number, the Fourier transform of the Cu K-edge in R space (FT-EXAFS) plot is fitted using the least-squares method. Based on the fitting results in Supplementary Fig. 13 and Supplementary Table 2, the coordination number (CN) of Cu-Cu in OD-Cu NWs is 9.1, which is much smaller than that of Cu NWs (10.3) and Cu foil (12), indicating that the lattice oxygen in OD-Cu NWs increases the unsaturation of the Cu-Cu coordination.

To understand the local atomic configuration and coordination structure of OD-Cu NWs, DFT-based AIMD simulations were performed. Based on the experimental characterization results, oxygen

residues with different contents are introduced to the lattice of Cu, and two models are built and named A-OD-Cu(111)-O$_{sub}$ and B-OD-Cu(111)-O$_{sub}$ (Supplementary Fig. 14). The Cu-Cu and Cu-O coordination on the two surfaces are investigated by Radius distribution functions (RDFs) at the time scale within 30 ps, as shown in Fig. 3f and Supplementary Fig. 15, respectively. Compared with A-OD-Cu(111)-O$_{sub}$ (Cu-Cu bond: 2.55 Å, CN: 10.12; Cu-O: 1.95 Å, CN$_{Cu-O}$: 0.35), B-OD-Cu(111)-O$_{sub}$ (Cu-Cu: 2.55 Å, CN: 8.9; Cu-O: 1.95 Å, CN$_{Cu-O}$:0.56) matches the static experimental characterization (XAFS) well in terms of bond length and coordination number. All these results clearly demonstrate that the oxygen residues inside the lattice of OD-Cu NWs exist in the form of Cu₄-O and alter the electronic and coordination structure of Cu.

**Assessment of the catalytic performance toward the acetonitrile reduction reaction**

To investigate the catalytic property toward the acetonitrile reduction reaction, electrochemical assessments on OD-Cu NWs and Cu NWs were carried out using a typical H-type cell. Figure 4a shows the linear sweep voltammetry (LSV) curves of Cu NWs and OD-Cu NWs in the Ar-saturated 1 M KOH electrolyte with/without 8 wt% acetonitrile additives. Apparently, both Cu NWs and OD-Cu NWs show significantly enhanced current density with 8 wt% acetonitrile. Meanwhile, the vigorous bursting of H₂ on the catalyst surface was suppressed with acetonitrile, indicating the preference of the acetonitrile catalytic reaction on copper-based catalysts (Supplementary Fig. 16). Moreover, the current density of OD-Cu NWs is much higher than that of Cu NWs in the whole studied potential region, suggesting much better kinetics on OD-Cu NWs. In addition, at high potential region (<−0.55 V vs RHE), significant bubble generation is still observed on Cu NWs electrode, while no obvious bubble on OD-Cu NWs is visually observed throughout the studied potential region, indicating that OD-Cu NWs

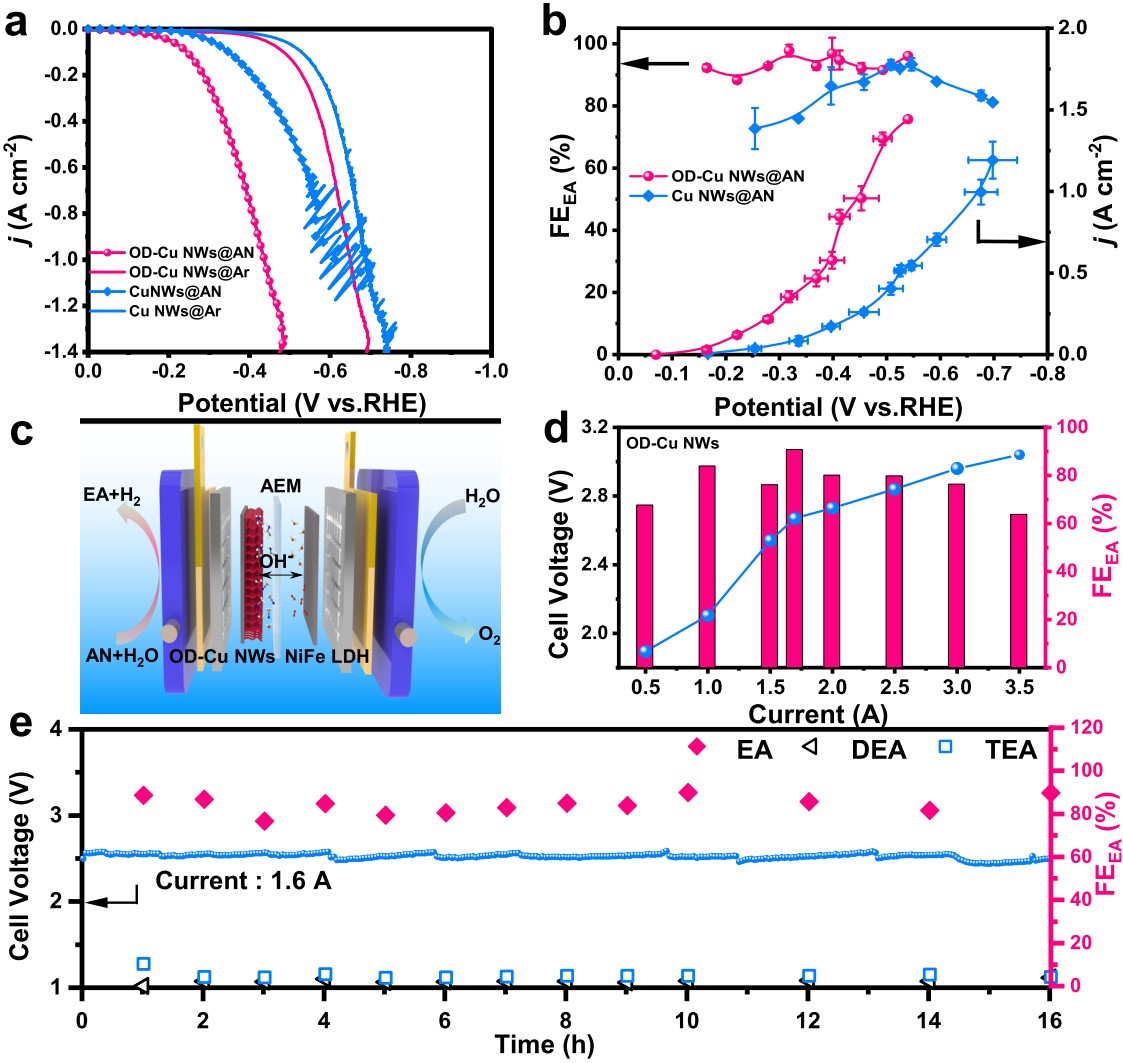

**Fig. 4 | Catalytic performances of OD-Cu and Cu nanowire samples in the acetonitrile reduction reaction. a** LSV plots of Cu NWs and OD-Cu NWs in argon saturated 1 M KOH aqueous solutions with or without the addition of AN (8 wt%). The scan rate was 10 mV s[-1]. **b** $FE_{EA}$ and current densities with different potentials on Cu NWs and OD-Cu NWs. **c** MEA device diagram. **d** AEM-MEA test of $FE_{EA}$ at different high currents in OD-Cu NWs. **e** Long-term durability and corresponding $FE_{EA}$ of OD-Cu NWs using AEM-MEA at a constant current of 1.6 A (in a 1 M KOH electrolyte containing 8 wt% AN). The error bars represent Fig. 4b based on triplicate measurements.

have a much wider catalytic potential window and better catalytic selectivity.

To quantitatively evaluate the ethylamine faradaic efficiency (FE) of the electrochemical acetonitrile reduction catalysis, [1]H nuclear magnetic resonance (NMR) and gas chromatography (GC) were used to detect the liquid and gas phase products (Supplementary Fig. 17), respectively. According to the [1]H-NMR and GC results recorded at different potentials shown in Fig. 4b and Supplementary Fig. 18, ethylamine was the major product on both OD-Cu NWs and Cu NWs. The FE of OD-Cu NWs is consistently maintained above 90% in the whole studied potential region, while Cu NWs showed apparently lower FE values, especially at high potential region due to the competition reaction of the HER. These results indicate that the lattice oxygen residual in OD-Cu NWs plays a key role in the catalytic activation of acetonitrile to form ethylamine. Furthermore, to better evaluate the production of ethylamine, the partial current density of ethylamine ($j_{EA}$) was calculated based on the FE and the total current density (Supplementary Fig. 18c, d). The OD-Cu NWs consistently deliver a substantially higher $j_{EA}$ than Cu NWs. At an applied potential of ~-0.5 V vs RHE, the $j_{EA}$ of OD-Cu NWs (1.21 A cm[-2]) is 3.2 times higher than that of Cu NWs ($j_{EA}$: 0.38 A cm[-2]), and the selectivity of ethylamine

for all amines was maintained at ~99% over the range of potentials tested, which indicates a drastically elevated acetonitrile reduction catalytic activity (Supplementary Fig. 18). Overall, with the assistance of lattice oxygen residues, OD-Cu NWs display not only better onset potential but also higher ethylamine $j_{EA}$ and FE than Cu NWs, which is also the state-of-the-art of the catalysts that have been reported (Supplementary Table 3). To explore the universality of the lattice oxygen-mediated electron tuning engineering, the developed strategy was expanded to other nitrile hydrogenations, including cyclopropanecarbonitrile (CPN), 3-hydroxypropionitrile (3-HPN), butyronitrile (BN), and pentanenitrile (PN) (Supplementary Figs. 19–23). For all the studied small molecular nitriles, the conversions to amines on OD-Cu NWs are always better than those on Cu NWs, with both higher current densities and faradic efficiencies, which further demonstrates that residual oxygen in OD-Cu NWs can efficiently manipulate the catalytic reduction of nitriles to primary amines.

To demonstrate its potential for industrial applications, the catalysts were further assembled in an anion-exchange membrane-membrane electrode assembly (AEM-MEA) and studied the ethylamine generation rate and long-term stability under large current conditions. Figure 4c illustrates the schematic diagram of the electrochemical

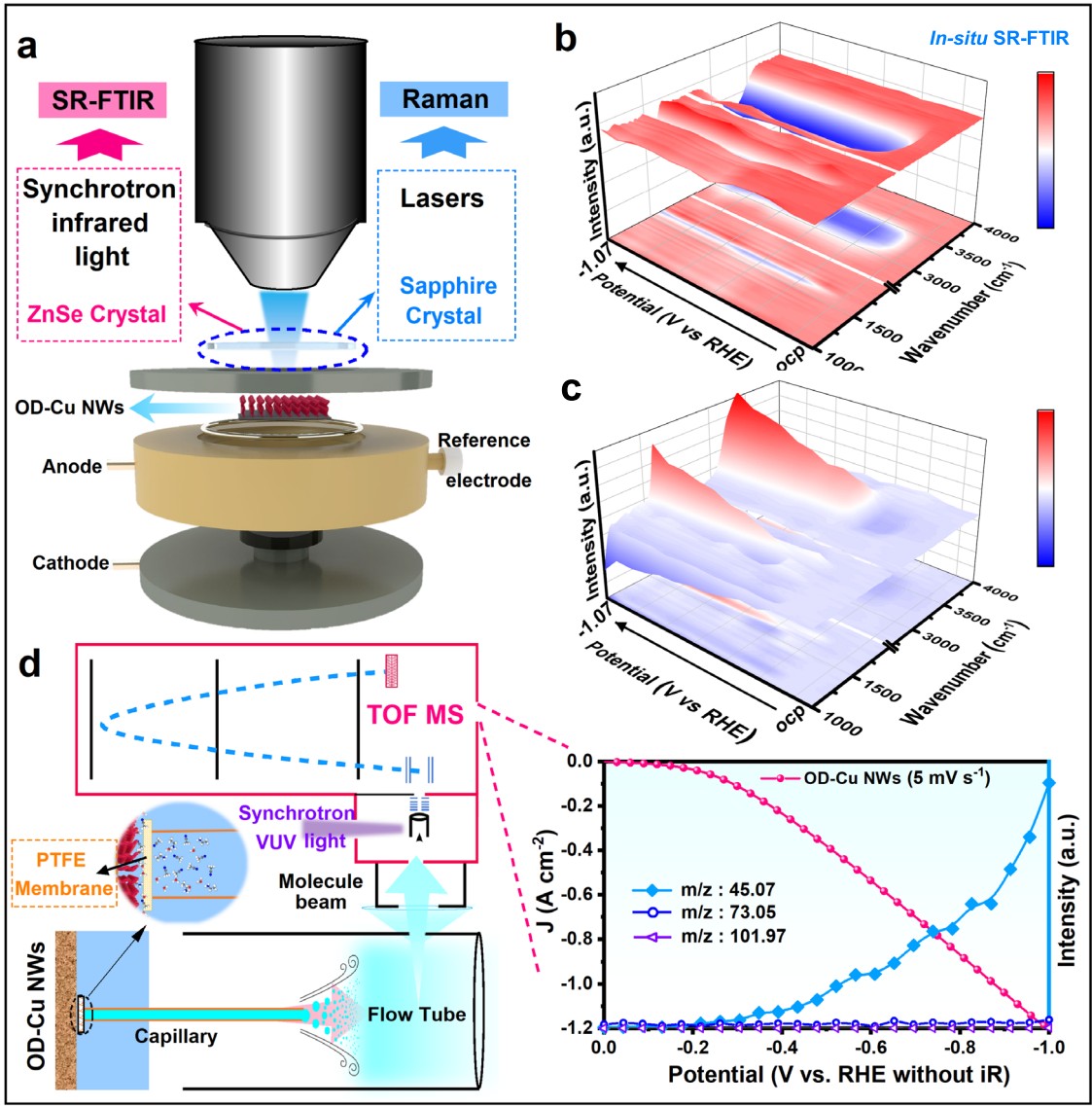

**Fig. 5 | In situ characterization of the ANRR mechanism. a** Schematic of in situ Raman and in situ SR-FTIR tests. **b, c** 3D plots of in situ SR-FTIR spectra of Cu NWs (**b**) and OD-Cu NWs (**c**) at different voltages. **d** Schematic of the SVUV-PIMS test and the SVUV-PIMS signal of the product recorded simultaneously with the LSV test.

reactor based on the AEM-MEA (Supplementary Figs. 24, 25), in which NiFe layered double hydroxide (NiFe LDH) loaded on nickel foam works as an anode for the oxygen evolution reaction (OER) and the OD-Cu NWs catalyst act as the cathode for acetonitrile reduction reaction. The polarization curves of OD-Cu NWs in 1 M KOH electrolyte with and without 8% acetonitrile are illustrated in Supplementary Fig. 26. In comparison with the high cell voltage of water splitting without the addition of acetonitrile, the cell voltages are significantly decreased after adding acetonitrile, demonstrating the occurrence of acetonitrile reduction reaction. Furthermore, the ethylamine faradaic efficiency (FE) in the AEM-MEA is also quantified, as shown in Fig. 4d. Notably, the FE of OD-Cu NWs is impressively maintained above 75% for EA formation over a wide current range from 1 to 3 A. Since durability is another important parameter for practical industrial applications in addition to high activity, a chronopotentiometry test is performed to evaluate the long-term operating stability of the OD-Cu NWs at a high current of 1.6 A (Fig. 4e). The OD-Cu NWs catalyst delivers stable voltage input and $FE_{EA}$ over 16 hours under continuous electrolysis. Meanwhile, after the stability test, the OD-Cu NWs can well maintain their initial nanowire morphology and fcc-Cu phase (Supplementary Fig. 27a, b). Furthermore, the Raman, core-level XPS and Cu L3M45M45

Auger spectra of OD-Cu NWs in Supplementary Fig. 27c–f prove the presence of lattice oxygen residues in OD-Cu NWs, which confirms the excellent structural stability. Overall, the high performance of the OD-Cu NWs using AEM-MEA opens up great opportunities in practical industrial applications for the electrochemical acetonitrile reduction reaction.

### In situ mechanism investigations of the acetonitrile reduction reaction over OD-Cu NWs

To gain the in-depth understanding of the electrochemical acetonitrile reduction reaction on OD-Cu NWs, in situ SR-FTIR and in situ Raman spectroscopy were carried out at different applied potentials (vs RHE without *iR* correction) in a 1 M KOH electrolyte containing 8 wt% acetonitrile. A schematic diagram of the in situ experimental device is shown in Fig. 5a, b and Supplementary Fig. 28a, b show the 3D and 2D in situ SR-FTIR spectra of Cu NWs collected by the reflection mode after background-subtraction. During the acetonitrile reduction reaction process, no visible band appears before −0.17 V, while at a potential of −0.17 V, the infrared bands of ν(N–H), ν(C–N) and ν(NH₂) stretching vibrations appear at ~1100 cm⁻¹, ~1400 cm⁻¹, and ~3000 cm⁻¹, confirming the generation of the corresponding reaction

intermediates during acetonitrile reduction (Supplementary Figs. 28)[15,41,42]. However, as the applied potential is further increased from −0.17 V to −1.07 V, the strong $\nu(NH_2)$ vibration signals for Cu NWs gradually increased and finally remained almost constant, verifying that the strong adsorption of EA* on metallic Cu results in large coverage of the $NH_2$-containing species and unfavorable desorption of EA* for EA formation. Impressively, after introducing lattice oxygen residues, when the applied potential is increased from −0.17 V to −1.07 V, no visible vibration of the -$NH_2$ band appears at ~3000 cm$^{-1}$ for OD-Cu NWs (Fig. 5c and Supplementary Fig. 28b), indicating that the binding strength of -$NH_2$ on OD-Cu NWs is moderate and that the desorption of -$NH_2$ is not significantly hindered.[41] Furthermore, in situ Raman spectroscopy of electrocatalytic ANRR on OD-Cu NWs at different potentials was performed. Similar to the SR-FTIR results, with increasing potential, the intensities of the Raman bands ascribed to the C−N stretching modes located at ~1340 cm$^{-1}$ and δ(OH) at 1623 cm$^{-1}$ gradually increase, while no Raman response attributed to $NH_2$ and NH is detected throughout the reaction (Supplementary Fig. 29). Meanwhile, the presence of Cu-O vibrational mode is maintained throughout the entire test duration, which confirms that the oxygen residuals on the OD-Cu surface are stable and favorable for the desorption of NH-containing species (Supplementary Fig. 30). In a word, the coverage of $NH_2$-containing species on the metallic Cu NWs surface limits the ANRR activity, especially at high current conditions, while introducing oxygen into the Cu lattice could facilitate -$NH_2$ desorption and benefit ANRR catalysis.

In addition, for more accurate real-time detection of the molecular products on the OD-Cu surface during the acetonitrile reduction reaction process, in situ SVUV-PIMS is further utilized, which provides time-resolved detection with a high sensitivity at a distance of approximately 10−100 μm from the electrode surface[16,43]. Figure 5d shows a schematic diagram of the device. Along with the electrochemical LSV, the signals of ethylamine (EA), diethylamine (DEA), and triethylamine (TEA) with m/z values of 45.016, 73.05, and 101.97 are observed with ionization energies of 11.6 eV (Supplementary Fig. 31)[44]. As shown in Fig. 5d, the onset potential for EA production was ~−0.17 V, while the signals of DEA and TEA are very limited, which is in accordance with the as-mentioned electrochemical and in situ spectroscopy results. Even at high current density, the signals of the possible byproducts of DEA and TEA are still negligible, further demonstrating the high selectivity of EA on the OD-Cu NWs catalyst.

## DFT calculations

To further elucidate the effect of the lattice oxygen in OD-Cu NWs on the enhanced reactivity and selectivity toward acetonitrile reduction, DFT calculations were performed. Based on the experimental characterizations (HRTEM, XPS, SRPES, and XAFS) and AIMD simulations, the B-OD-Cu(111)-$O_{sub}$ surface (denoted as B-OD-Cu) was constructed to simulate OD-Cu NWs (Supplementary Fig. 14), while the Cu(111) surface was also analyzed for comparison (Supplementary Fig. 32). Notably, the presence of lattice oxygen residual lengthens the surface Cu-Cu bond of B-OD-Cu compared with Cu(111) (Supplementary Fig. 33), which is in accordance with the HRTEM and HADDF-STEM results. In addition, based on the Bader charge analysis, $Cu^{δ+}$ exists mainly inside B-OD-Cu, while surface Cu atoms are only slightly positively charged, similar to the valence state detected by the SRPES (Fig. 6a). Furthermore, the stability of the lattice oxygen residual in the model is demonstrated by the AIMD simulations in Fig. 6b and Supplementary Fig. 34, where the lattice oxygen residual is stable within a 30 ps simulation time at 300 K based on the temperature, energy versus time plot, the root mean square deviation (RMSD) and tracking of the Cu-O bond and Cu-Cu length change.

The effect of residual oxygen on the intrinsic properties of copper was investigated. As shown by the electron localized function (ELF) in Fig. 6c, the electrons on the B-OD-Cu surface are more localized than

that on Cu(111), which implies that the valence electrons of B-OD-Cu have difficulty moving randomly on the surface. In addition, based on crystal orbital Hamilton population (COHP) analysis, electron localization might originate from electron transfer from the Cu-Cu 4 s @ 4 p bonding state to the Cu-O 3 d-2 p antibonding state near the Fermi energy level (Fig. 6d). Overall, residual oxygen inside the lattice, as the electron acceptor, draws charge from the s @ p band of the OD-Cu surface and localizes surface electrons, which could lead to the weakened adsorption of reaction intermediates.

Since the surface adsorption behavior of the catalyst is crucial to determine the catalytic performance, the corresponding adsorption energies of the key intermediates ($CH_3CN^*$, $CH_3CHN^*$, $CH_3CNH^*$, $CH_3CHNH^*$, $CH_3CH_2N^*$, $CH_3CNH_2^*$, $CH_3CH_2NH^*$, $CH_3CHNH_2^*$, $CH_3CH_2NH_2^*$ and $H_2O^*$) involved in acetonitrile reduction are compared. On the B-OD-Cu surface, the confined lattice oxygen can effectively modulate the adsorption behavior of intermediate species, resulting in overall weakened adsorption of all the intermediate species compared with Cu(111), especially for the key adsorption species ($H_2O$, $CH_3CN$, and $CH_3CH_2NH_2$) (Supplementary Fig. 35). Furthermore, the effect of different amounts of residual lattice oxygen on the adsorption of key adsorption species ($H_2O^*$, $CH_3CN^*$ and $CH_3CH_2NH_2^*$) was further explored. Figure 6e illustrates the adsorption energies of the key adsorption species on four surfaces with different residual oxygen contents, including Cu(111), A-OD-Cu, B-OD-Cu, and CuO(111), with oxygen atomic ratios of 0, 5.8%, 11%, and 50%, respectively. The A-OD-Cu model is determined in the same way as B-OD-Cu (Supplementary Fig. 36). Notably, the $CH_3CH_2NH_2$ adsorption on Cu(111) is quite strong, as reflected by a large adsorption energy of −1.08 eV. Interestingly, with the addition of small amounts of O atoms (5.8% and 11%) into the Cu NWs catalyst, $CH_3CH_2NH_2$ adsorption is significantly decreased, which is beneficial for $CH_3CH_2NH_2$ desorption from the catalyst surface. Meanwhile, $H_2O$ adsorptions on both A-OD-Cu and B-OD-Cu surfaces are weakened compared with those on Cu(111) and CuO(111), suggesting that the residual oxygen as an electron acceptor can effectively modulate the adsorption behavior of intermediate species involved in the ANRR by manipulating the localization of electrons on the surface.

In addition to structural information, the reaction mechanism is finally investigated. Based on the calculated infrared spectra (IR) of intermediate species in the electroreduction of acetonitrile to ethylamine on Cu(111) and B-OD-Cu(111) surfaces, that ethylamine adsorbed on Cu(111) is the potential-determining step due to its strong adsorption, while on B-OD-Cu(111), the PDS is shifted to the proton-coupled electron transfer electrochemical process of $CH_3CHNH^*$ to $CH_3CH_2NH^*$, which is consistent with the SR-FTIR results (Supplementary Fig. 37). Furthermore, Supplementary Fig. 38 and Fig. 6f display the possible formation pathways of EA on the Cu(111) and B-OD-Cu surfaces. For Cu(111), starting from $CH_3CN$ adsorption, the optimal formation route of ethylamine formation is $CH_3CN^* \rightarrow CH_3CHN^* \rightarrow CH_3CHNH^* \rightarrow CH_3CHNH_2^* \rightarrow CH_3CH_2NH_2^* \rightarrow CH_3CH_2NH_2$ (aq). It is clear that due to the strong adsorption of $CH_3CH_2NH_2^*$, the desorption of $CH_3CH_2NH_2^*$ is the potential-determining step (PDS) of the reaction, requiring a high energy barrier of 0.69 eV. In comparison, the optimal reaction pathway on B-OD-Cu is $CH_3CN^* \rightarrow CH_3CHN^* \rightarrow CH_3CHNH^* \rightarrow CH_3CH_2NH^* \rightarrow CH_3CH_2NH_2^* \rightarrow CH_3CH_2NH_2$ (aq). Due to the weakened $CH_3CH_2NH_2^*$ adsorption, the desorption of $CH_3CH_2NH_2^*$ is favored. Thus, the PDS is switched from the $CH_3CH_2NH_2^*$ desorption into a proton-coupled electron transfer process of $CH_3CHNH^* + H^+ + e \rightarrow CH_3CH_2NH^*$ with a small energy barrier of only 0.22 eV. In addition, the energy barrier of the HER is also higher and up to 0.34 eV with the Volmer-Heyrovsky mechanism, further indicating the suppression of the HER reaction on the B-OD-Cu surface, in line with the results of our electrochemical tests. In addition, both experimental and theoretical investigations indicate that the variation in the CN of Cu catalysts and the dynamically adsorbed-oxygen-containing species on their surfaces

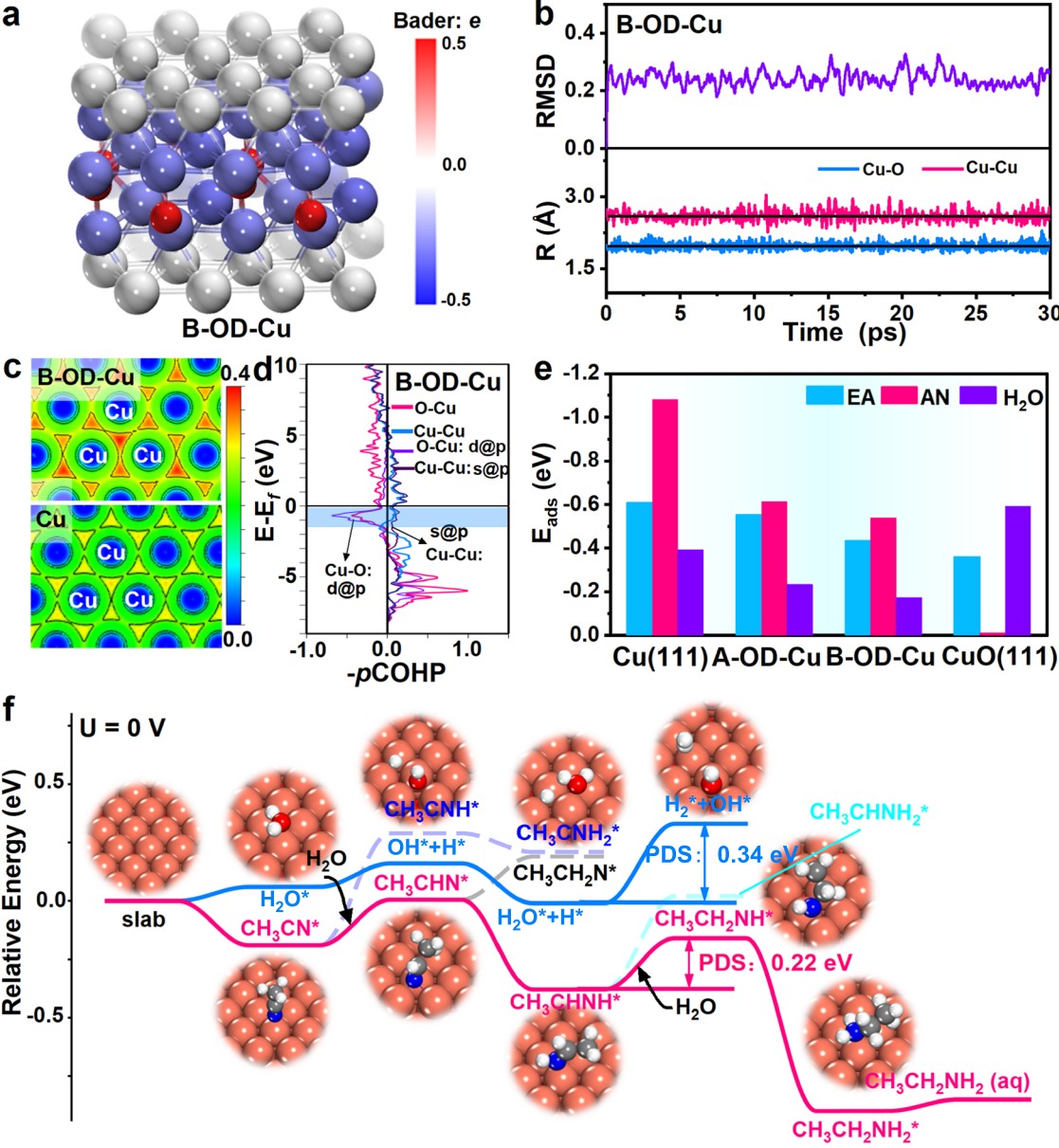

**Fig. 6 | DFT calculations. a** Bader charge of the B-OD-Cu surface. **b** The RMSD and the tracking of Cu-O bond and Cu-Cu bond length changes at 300 K in the AIMD simulations. **c** ELF maps of B-OD-Cu and Cu(111), respectively. **d** COHP maps of Cu-Cu bonds and Cu-O bonds in B-OD-Cu. **e** The adsorption energies ($\Delta E_{ads}$/eV) of AN, EA, and H₂O on Cu(111), A-OD-Cu, B-OD-Cu, and CuO(111), with oxygen contents corresponding to 0, 5.8%, 11%, and 50%, respectively. **f** Potential energy diagram of all possible pathways and corresponding geometric structures for the generation of EA by AN electroreduction on the B-OD-Cu surface at 0 V (vs RHE).

might not be the decisive factors for the efficient conversion of AN to EA (Supplementary Figs. 39–41). Thus, the localized surface electron distribution induced by lattice oxygen residuals is the key to decreasing the ethylamine binding energy on the Cu metal surface.

## Discussion

In summary, we have demonstrated that lattice oxygen residue-mediated electron modulation inside metallic Cu can intrinsically promote the electroreduction of acetonitrile. XPS, XAS, and AIMD results reveal that the oxygen residues inside the lattices of OD-Cu NWs exist in the form of Cu₄-O, which consequently alters the electronic and coordination structures of Cu. Impressively, the prepared OD-Cu NWs achieve a maximum FE$_{EA}$ of ~97.8% at −0.32 V (vs RHE) while maintaining a FE$_{EA}$ of 91% at a large current density of more than 1.0 A cm$^{-2}$ at ~−0.44 V vs RHE. A series of advanced in situ characterizations (in situ Raman spectroscopy, in situ SR-FTIR, and in situ SVUV-

PIMS) further indicates that the presence of residual oxygen in OD-Cu NWS effectively weakens the adsorption of EA on the surface and thus leads to selective formation of the EA product. Meanwhile, DFT calculations indicate that the weakened adsorption of EA and improved catalytic activity are attributed to the confined electron effect induced by residual lattice oxygen. This work layout the modulation principles underlying the improved catalytic performance, which could provide valuable insights for the design of electrocatalysts toward small organic molecule catalysis.

## Methods

### Synthesis of CuO NWs precursors

In a typical procedure, 2 × 3 cm² of Cu foam was washed with alcohol, concentrated HCl, and deionized water to clean the surface. Next, the washed Cu foam was soaked in 0.1 M (NH₄)₂S₂O₈ with 1 M NaOH for 30 min at ~5 °C to oxidize Cu to Cu(OH)₂. The synthesized Cu(OH)₂ was

then washed with deionized water and dried in an oven at 60 °C for 1 h. Subsequently, $Cu(OH)_2$ was annealed in flowing argon at 180 °C for 1 h to convert it into CuO NWs.

## Synthesis of OD-Cu NWs

In situ electrochemical reduction of the annealed CuO NWs was performed using the chronoamperometric method at −0.37 V vs RHE for 300 s to obtain metallic Cu nanowires with limited oxygen residues (OD-Cu NWs) (Supplementary Fig. 1).

## Synthesis of Cu NWs

Cu NWs were synthesized by annealing CuO NWs for 3 h at 350 °C in a 3% $H_2$/Ar flow atmosphere.

## Characterization

The X-ray diffraction patterns of the samples were obtained on a Rigaku Miniflex- 600 operating at 40 kV voltage and 15 mA current with Cu Kα radiation (λ = 0.15406 nm). Scanning electron microscopy (SEM, JEOL-JSM-6700F, 5 kV of accelerating voltage) and transmission electron microscopy (TEM, Hitachi H7650, 100 kV of accelerating voltage) were employed to collect morphological and microstructural information. Raman spectra were recorded on a Lab RAM HR JY-Evolution microscope using a 532 nm argon ion laser. X-ray photoelectron spectroscopy (XPS) was collected on a scanning X-ray microprobe (PHI 5000 Versasa, ULAC-PHI, Inc.) by Al Ka radiation, and the C 1 s peak located at 284.8 eV was used as a standard. The high-resolution TEM, and HAADF-STEM were recorded by a JEOL JEM-ARM200F TEM/STEM with a spherical aberration corrector working at 200 kV. The TPR-$H_2$ experiment was carried out on an Auto Chem II 2920. The synchrotron radiation photoemission spectroscopy (SRPES) spectra and soft X-ray absorption spectroscopy (XAS) were measured at the BL10B end-station in the National Synchrotron Radiation Laboratory (NSRL) of Hefei.

## X-ray absorption fine structure (XAFS) spectroscopy measurement and data analysis

The as-prepared samples were characterized by XAFS spectroscopy to provide a description of the local coordination environment around the copper atoms. Cu K-edge XAFS spectroscopy was gathered at the beamlines BL14W1 and BL11B in the Shanghai Synchrotron Radiation Facility (SSRF). The storage ring operates at an energy of 3.5 GeV and a current of ~240 mA. The white X-rays were monochromated with a Si(111) bicrystal monochromator, and the energy was calibrated with Cu foil. Cu standards and prepared samples were collected in transmissive mode at room temperature. The data processing was performed using the software Demeter[45].

## Notes

Considering the susceptibility of Cu NWs and OD-Cu NWs to oxidation, to protect Cu NWs and OD-Cu NWs from oxidation, both Cu NWs obtained from $H_2$ treatment and OD-Cu NWs obtained from electrochemical reduction were rapidly stored in an oxygen/water-free an argon-filled glove box. For characterization, samples were placed in argon-saturated ethanol that was restored in an oxygen/water-free and argon-filled glove box. After sonication, the powders were collected by natural drying in the glove box.

## Electrochemical measurements

Electrochemical measurements were performed at ambient temperature and pressure in a customized gastight H-type cell separated. A CHI 760e electrochemical workstation was employed to record the electrochemical response. A typical three-electrode cell was employed with a piece of platinum wire and a Ag/AgCl electrode (KCl sat.) serving as the counter electrode and reference electrode, respectively. All potentials measured were calibrated to the RHE scale as follows:

$E_{RHE} = E_{Ag/AgCl} + 0.197 V + 0.0591 V × pH$, with 80 % ohmic resistance correction applied in all the measurements. Chronoamperometric electrolysis was conducted at each potential for a total time of 600 s. The liquid products were analyzed by 1H NMR spectroscopy. 1H NMR spectra measured with water suppression using a presaturation method were collected on a Bruker 400 MHz spectrometer to test the liquid products. Typically, 500 μL of electrolyte after ANRR electrolysis was mixed with 200 μL of $D_2O$ containing 14 ppm (m/m) dimethyl sulfoxide as the internal standard. The same spectral acquisition parameters were used for all measurements to ensure complete relaxation and quantification. The faradaic efficiencies (FEs) of the liquid products were calculated using the following equation:

$$FE\,(\%) = \frac{eFn}{Q} \times 100\,\% \tag{1}$$

where $e$ is the number of electrons transferred, $F$ is the Faraday constant, $n$ is the amount of product in moles, $Q$ is the charge.

The GC system was equipped with hayesep D column with Ar (Praxair, 5.0 Ultrahigh purity) flowing as a carrier gas and 5 A columns connected to a thermal conductivity detector and a flame ionization detector. The faradaic efficiencies (FEs) of the gas products were calculated using the following equation:

$$FE\,(\%) = \frac{eCfFP}{RTI} \times 100\,\% \tag{2}$$

where $e$ is the number of electrons transferred to $H_2$ formation (2), $F$ is the Faraday constant (96485 C/mol), $C$ is the measured concentration of the product by GC (in ppm), $f$ is the gas flow rate (ml/s), $P$ is the pressure ($1.01 × 10^5$ Pa), I is the imposed current (in $A$).

## AEM-MEA measurements

The membrane electrode assembly (MEA) used in the experiments consists of a piece of 2 cm × 2 cm × 0.4 cm NiFe LDH anode (NiFe LDH was synthesized based on the previously reported method[46], an AEM, and a 2 cm × 2 cm × 0.2 cm OD-Cu NWs cathode (Supplementary Fig. 18). The AEM (NPPO-2QA-x polymer, which is a modified PPO polymer backbone polymerization with 1,3-bis(trimethylammonium-bromide-methyl)−5-(prop-2-ynyloxy) benzene (TABB) and azide modifications, the thickness of 50 μm) was treated with 1.0 M KOH solution to remove possible additives prior to use[47]. Electroreduction of acetonitrile to ethylamine was tested at room temperature with an argon-saturated 1 M KOH aqueous electrolyte containing 8 wt% acetonitrile at a flow rate of 10 mL/min at the cathode and a 1 M KOH electrolyte at the anode. Chronopotentiometry experiments were conducted to evaluate the acetonitrile electroreduction performance using an ITECH Auto Range DC power supply. For each data point, after the electrolysis reached a stable state, 10 mL of the liquid product was collected and the corresponding time was recorded. The Faraday efficiency (FE) of the liquid product is calculated in accordance with the three-electrode test.

## In situ measurements

In situ SR-FTIR measurements were performed on the infrared beamline BL01B of the National Synchrotron Radiation Laboratory (NSRL, Hefei, China) via an in situ reflection infrared setup with a ZnSe crystal as the infrared transmission window. The in situ test was carried out in argon saturated 1 M KOH aqueous solutions with/without the addition of acetonitrile (8 wt%) solution. During the test, the catalyst electrode was pressed tightly against the ZnSe crystal window with a micron gap to reduce the loss of IR light[41]. To ensure the quality of the obtained SR-FTIR spectra, we used a reflection mode with a perpendicular incidence of IR light and tested in the range of 600–4000 cm$^{-1}$. Each IR absorption spectrum was obtained by averaging 128 scans.

In situ Raman spectroscopy was measured in an in situ Raman setup with 1 M KOH containing 8 wt% acetonitrile poured into an electrolytic cell as the electrolyte, using platinum wire, Ag/AgCl (KCl saturated) and OD-Cu NWs loaded on a glassy carbon electrode as the counter, reference, and working electrodes. All Raman spectra were collected with a constant potential applied to the working electrodes under an excitation laser source of 532 nm.

In situ SVUV-PIMS experiments were conducted at the Atomic & Molecular Physics Beamline (BL09U) of the National Synchrotron Radiation Laboratory in Hefei, China. The determination of gaseous and volatile reaction products was performed by on-line SVUV-PIMS in parallel with electrochemical measurements. The reaction products were collected through a PEEK capillary (inner diameter 0.2 mm) covered by a hydrophobic PTFE membrane. The capillary was in contact with the cathode surface. The PTFE membrane is designed to prevent the introduction of aqueous electrolytes while allowing volatile gaseous products to enter the vacuum chamber via the differential pumps. Product samples are ionized at an ionization energy of 11.6 eV.

## Computational details

All density functional theory (DFT) calculations employ the plane-wave basis functions to expand the atomic core and valence electrons with a cutoff kinetic energy of 400 eV, which are implemented in the Vienna Ab initio Simulation Package (VASP 5.4.4)[48,49]. The core−valence interactions were described by the projector-augmented wave (PAW) method[50]. Spin-polarized Kohn-Sham formalism with Perdew–Burke–Ernzerhof (PBE) flavor of generalized gradient approximation was employed[51]. The convergence criteria for energy and force were set to $1.0 \times 10^{-5}$ eV and $0.01$ eV Å$^{-1}$ for all geometric optimizations, respectively. Meanwhile, the method of DFT-D3 with the Becke-Johnson damping function was employed to correct the adsorption energy[52]. Monkhorst−Pack $(3 \times 3 \times 1)$ Γ-centered grid sampling for the Brillouin zone was used for surface geometry optimization, and Monkhorst−Pack $(5 \times 5 \times 5)$ Γ-centered grid sampling was used for bulk geometry optimization[53]. The finite difference method is used to calculate the vibrational modes of surface-adsorbed species and thus obtain the corresponding zero-point energy (ZPE), enthalpy, and entropy. Meanwhile, the infrared spectra (IR) of all intermediate species involved in the electroreduction of acetonitrile to ethylamine in their most stable adsorption configurations were obtained using the symmetry-based density functional perturbation theory (DFPT) method.

Furthermore, we use VASPsol with water as the solvent, a software package that incorporates solvation into the self-consistent continuum model of VASP, for all calculations of the solventized calibration of the free energy[54]. Polarizable continuum models (PCMs)[55] were employed to model the solvation effect. The solvent energy is was evaluated using the following expressions:

$$E_{\text{solv}} = E_{\text{vaspsol}} - E_{\text{vasp}} \tag{3}$$

where $E_{\text{vaspsol}}$ refers to the energy calculated by VASPsol and $E_{\text{vasp}}$ stands for the energy calculated by VASP.

Free energy corrections for all species involved in the ANRR reaction were performed by VASPKIT[56]. For free gas molecules, the ideal gas approximation was assumed. For adsorbates, the contribution of all degrees of freedom to the free energy is considered as vibrations under the harmonic approximation. The reaction free energy of each step is calculated by:

$$\Delta G = \Delta E + \Delta ZPE - T\Delta S + E_{\text{solv}} \tag{4}$$

where ΔE refers to the total energy, ΔZPE refers to the zero-point energy correction, and ΔS refers to the vibration entropy change.

The adsorption energy of adsorbate A was defined as:

$$E_{\text{ads}} = E_{\text{slab}+A} - (E_{\text{slab}} + E_A) + \Delta ZPE + E_{\text{solv}} \tag{5}$$

where $E_{\text{slab}+A}$ is the total energy for the slabs with adsorbate on the surface, $E_{\text{slab}}$ is the total energy of the slab, and $E_A$ is the total energy of free surface adsorbate A. Thus, the more negative adsorption energy, the stronger interaction between adsorbate and catalyst surface.

Ab initio molecular dynamics simulations (AIMD) were carried out to assess the thermodynamic stability of OD-Cu NWs under the Born–Oppenheimer approximation. At the onset of the MD simulations, the initial temperature of the OD-Cu NWs sample was 100 K according to the Boltzmann distribution. The sample was then heated to the required temperature (300 K) by a 5 ps velocity scale and then equilibrated with a Nosé thermostat for 30 ps at equilibrium temperature with a constant volume[57]. The time step was chosen to be 1 fs, and the integration of Newton's equation was based on the Verlet algorithm implemented in VASP.

## Surface models

Based on XRD and HAADF-STEM characterization results, the catalyst surface models were based on the (111) oriented surface of Cu crystal structure. Furthermore, fine structural analyses (SRPES and XAS) indicate the oxygen residues inside the lattice of OD-Cu NWs exist in the form of Cu$_4$-O. Oxygen residues with different contents are introduced to the lattice of Cu and two models are built and named as A-OD-Cu(111)-O$_{\text{sub}}$ and B-OD-Cu(111)-O$_{\text{sub}}$. For the A-OD-Cu(111)-O$_{\text{sub}}$ surface, a four-layer $p(2\sqrt{2} \times \sqrt{2})$ Cu (111) with limited oxygen residual was built, which includes 64 copper atoms and 4 oxygen atoms (Supplementary Fig. 14) with an oxygen atomic ratio of 5.8%. For B-OD-Cu(111)-O$_{\text{sub}}$ surface, a four-layer $p(2\sqrt{2} \times \sqrt{2})$ Cu (111) with limited oxygen residual was built, including 64 copper atoms and 8 oxygen atoms (Supplementary Fig. 14) with an oxygen atomic ratio of 11%. In addition, a four-layer $p(4 \times 4)$ Cu(111) (includes 64 copper atoms) and a four-layer $p(3 \times 2)$ CuO(111) (includes 48 copper atoms and 48 oxygen atoms) surface slabs were constructed for comparison, where the oxygen atom ratios were 0% and 50%, respectively. Furthermore, the effects of unsaturated surface ligands by constructing high index surfaces of copper (Cu (100), Cu (110), and Cu (211)) and unsaturated ligands by creating copper vacancies under the surface (Cu (111)-d) on the reduction of acetonitrile were investigated. For unsaturated surface ligand model builds, four layers of $p(4 \times 4)$ Cu(100) (including 64 Cu), six layers of $p(3 \times 4)$ Cu(110) (including 72 Cu, the bottom two layers were fixed) and ten layers of $p(2 \times 4)$ Cu (211) (including 80 Cu, the bottom four layers were fixed) were built. For the unsaturated ligand model construction, a four-layer $p(4 \times 4)$ Cu(111) (including 62 Cu atoms, with two Cu atoms missing from the subsurface layer) was built (Cu (111)-d) (Supplementary Fig. 40). Furthermore, the repeated slabs were separated from their neighboring images by a 15 Å width vacuum in the direction perpendicular to the surface. In all calculations, unless otherwise stated, the bottom layer was fixed to its buck positions and the other layers with the adsorbed species were allowed to relax.

## Data availability

The data that support the findings detailed in this study are available in the Article and its Supplementary Information or from the corresponding authors upon reasonable request.

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

## Acknowledgements

The authors thank the financial support of the National Key Research and Development Program of China (2021YFA1500400), the financial support of the Natural Science Fund of China (22175163), the Fundamental Research Funds for the Central Universities (WK2060000016), Youth Innovation Promotion Association of the Chinese Academy of Science (2017483), the Plan for Anhui Major Provincial Science & Technology Project (Grants 202103a05020015 and 2021d05050006) and the Collaborative Innovation program of Hefei Science Center, CAS. We also acknowledge the Shanghai Synchrotron Radiation Facility (BL14W1 and BL11B, SSRF) and the Hefei National Synchrotron Radiation Laboratory (BLO1B, BLO9U, and BL10B, NSRL) for SR-FTIR, SVUV-PIMS, SRPES, and XAS characterizations. The numerical calculations in this paper were performed in the Supercomputing Center of University of Science and Technology of China.

## Author contributions

Conceptualization: C.W., Y.F., and G.W. Data curation: C.W. Formal analysis: C.W., and Y.F. Investigation: C.W. Methodology: C.W., Y.F., B.L., C.T., B.D., X.Y., Z.B., and Z.W. Funding acquisition, project administration, resources, and supervision: G.W. Validation: C.W. and Y.F. Visualization: C.W. Writing—original draft: C.W. and Y.F. Writing—review and editing: C.W., Y.F., J.L., Y.Q., and G.W. All authors discussed and analyzed the data.

## Competing interests

The authors declare no competing interests.
