## [Peer Review File · Nature Communications]

REVIEWER COMMENTS

Reviewer #1 (Remarks to the Author):

In this work, Wang and co-workers describe the promoted electrosynthesis of ethylamine from acetonitrile using OD-Cu NWs with abundant lattice oxygen. By a series of in situ characterizations, the authors confirm the presence of residual lattice oxygen in OD-Cu NWs, which plays a vital role in boosting the electroreduction of acetonitrile by facilitating the desorption of ethylamine. Although the discussion about lattice oxygen in materials is important and the characterizations are completed, I do not think this work can be published in Nature Communications on the consideration of novelty and the following points:

1. The same material and the same reaction have been reported by Lv (Chem Catalysis 2021, 1, 393-406) and Jiao (Nat. Commun. 2021, 12, 1949), respectively. The FE in this work is ~97.8%, which is only slightly higher than the reported work (~94.6%, Nat. Commun. 2021, 12, 1949). So, compared with the above-mentioned catalytic system, I could not find the innovation and importance of this one and is only an addition to the field. What's more, the authors said, "...provide valuable insights for the design of electrocatalysts toward small organic molecule catalysis". Questions like what happens to the other types of small molecule nitriles must be addressed.
2. Much of the modeling and interpretation of the catalytic performance could be overturned if the coordination number (CN) of Cu-Cu, rather than the residual lattice oxygen, is proven to cause the differences in electrochemical performance between OD-Cu NWs and Cu NWs. The lattice oxygen, which is bonded with Cu as depicted in the inner graphic of Fig. 3b, is not detected either in OD-Cu NWs or Cu NWs (Fig. 3e). The significant difference between them is the CN of Cu-Cu (9.1 vs 10.3). The author should better explain about this point.
3. The results of Supplementary Table 1 and 2 are in conflict. OD-Cu NWs is well fitted to 27.8% Cu₂O with Cu₄-O coordination and 72.2% Cu foil in Supplementary Table 1, but the fitting results in Supplementary Table 2 do not support the exit of Cu-O coordination of OD-Cu NWs.
4. The EXAFS fitting results in Supplementary Table 2 show the R of Cu-Cu is 2.54 Å for both OD-Cu NWs and Cu NWs. However, the XRD, HRTEM, and HAADF-STEM results in Fig. 2 indicate an obvious lattice expansion in OD-Cu NWs as compared to Cu NWs. These results seem also contradictory.
5. The comparison should be more accurate if the O K-edge NEXAFS of Cu NWs is provided in Fig. 3b.

6. The characterizations of the catalysts after electrocatalytic hydrogenation of nitriles should be studied to determine if they have changed during the course.

7. The OD-Cu NWs with lattice oxygen are prepared at -0.37 V vs. RHE, which is the reduction potential of Cu₂O to Cu. However, the applied potential for the hydrogenation of nitriles is more negative than -0.37 V (Fig. 4b). And, what are the cathodic potentials when conducting the hydrogenation reaction of nitriles at the current density of more than 1.5 A in Fig. 4d? At such negative potentials or high current densities, can the lattice oxygen in OD-Cu stably exist? Or in fact, the real catalyst for the hydrogenation of nitriles is not the same as the OD-Cu NWs discussed in this work.

8. The adsorption modes, such as adsorption via only the N atom or both the C and N atoms, of various molecules should be optimized and illustrated because different adsorption models lead to different shifts in FTIR and Raman spectra (J. Phys. Chem. C 2019, 123, 2300-2313). The simulated results should be compared with the experiment data to exactly determine the adsorption of the reaction species, hence a more accurate reaction mechanism.

9. The $\nu(\text{NH}_2)$ vibration cannot be distinguished because it is overlapped with the $\nu(\text{OH})$ vibration in Supplementary Fig. 20a.

10. Moreover, there are some mistakes in the main text, such as the “desorption” in line 76 should be adsorption, there is an extra “b” in line 231, and “jEA” in line 243 should be italicized.

11. Does this electrocatalytic hydrogenation method apply to other nitriles? I'm just giving a suggestion.

Reviewer #2 (Remarks to the Author):

In this manuscript, wei and co-workers reveal that residual lattice oxygen in oxide-derived Cu nanowires (OD-Cu NWs) plays vital roles in boosting the acetonitrile electroreduction efficiency. A series of in situ characterization techniques were applied to understand the role of oxygen residual in the catalytic process. Theoretical calculations suggest that oxygen residues act as electron acceptors to confine the free electron flow on the Cu surface, consequently improving the desorption of ethylamine. Overall, this work is interesting and well presented, and the understanding on the effect of oxygen residual is also

systematical and comprehensive by using various techniques. Before acceptance, necessary revisions are required:

1. The authors claim that oxide-derived copper has better performance due to the residue oxygen in copper lattice. However, Cu is an active metal especially unsaturated Cu after electrochemical reduction which could be easily oxidized. In situ characterizations should be added to exclude the residual oxygen which is not from the external environment.
2. In the DFT calculation, two models are built to simulate the OD-NW. The authors claim that the second model is chosen according to the experimental characterization results but did not explain the reason. And why only underlayer oxygen is considered and surface oxygen is not considered?
3. The O K-edge NEXAFS of Cu NWs should be provided for comparison. In addition, detailed valence states of Cu should be calculated in order to better understand the effect of O residues on the electronic structure of Cu.
4. The authors should specify which kind of AEM is used in this study and how is the stability of AEM during the electrolysis.
5. Necessary information should be clearly presented. For example, the attribution of each peak in Supplementary Fig. 21.

Reviewer #3 (Remarks to the Author):

attached

The manuscript entitled “Lattice oxygen mediated electron tuning promotes electrochemical hydrogenation of acetonitrile on copper catalysts” by Wei, et al. reported a study of the electrocatalytic nitriles hydrogenation primary amines on Cu-based catalysts. In particular, the authors attributed the observed high acetonitrile electroreduction efficiency to the residual lattice oxygen in oxide-derived Cu nanowires (OD-Cu NWs). They identified that the “oxygen residues, in the form of Cu₄-O configuration, act as electron acceptors to confine the free electron flow on the Cu surface, consequently improving the kinetics of nitriles hydrogenation catalysis.” A key claim of the manuscript is that the “oxygen residues” promote “electron confining capability of copper atoms” and weakens the amine binding strength. On the other hand, desorption of the amine is a nonelectrochemical step and the characterizations do not provide convincing support of the promotion effect.

In summary, although this manuscript reported some improvements over the previous publications, the overall progress is incremental and does not warrant its publication in Nature Communication.

Specific comments:

1. The manuscript quoted a total current density of 50 mA cm⁻² but failed to point out the partial EA current density is ~0.85 A cm⁻² for ref. 17.
2. The manuscript did not report hydrogen production in the experimental results although proton reduction was included in the mechanistic study. What is the EA selectivity?
3. The authors determined that the subsurface oxygen in the form of “B-OD-Cu” is the active configuration for EA formation. Is this model consistent with the oxygen concentration in the OD-Cu NWs? The authors quote a CN of 4.4 for B-OD-Cu model. Is this local coordination number for the O atom buried in the subsurface hollow site? How does this compare with the averaged O-Cu coordination number from XAFS? Alternatively, will the dynamically formed oxygen-containing species such as OH and surface oxygen contribute to EA formation activity?
4. English writing needs improvement. The manuscript should be carefully proofread.

Point-by-point response to the referees' comments

We sincerely thank the referees for their thorough review and valuable comments, which have significantly improved the quality of our manuscript. We also extend our gratitude to the editor for the offered opportunity to revise the manuscript. In response to the referees' comments, we have revised the manuscript and highlighted the changes in yellow. The point-by-point response to the referees' comments are outlined below.

Reviewer #1 (Remarks to the Author): *In this work, Wang and co-workers describe the promoted electrocatalysis of ethylamine from acetonitrile using OD-Cu NWs with abundant lattice oxygen. By a series of in situ characterizations, the authors confirm the presence of residual lattice oxygen in OD-Cu NWS, which plays a vital role in boosting the electroreduction of acetonitrile by facilitating the desorption of ethylamine. Although the discussion about lattice oxygen in materials is important and the characterizations are completed, I do not think this work can be published in Nature Communications on the consideration of novelty and the following points:*

Response: We appreciate the reviewer's positive feedback on our work regarding the importance of lattice oxygen in materials and the completeness of our characterizations. To address the other comments on the novelty, we have prepared a point-by-point response, which are presented below.

1. *The same material and the same reaction have been reported by Lv (Chem Catalysis 2021, 1, 393-406) and Jiao (Nat. Commun. 2021, 12, 1949), respectively. The FE in this work is ~97.8%, which is only slightly higher than the reported work (~94.6%, Nat. Commun. 2021, 12, 1949). So, compared with the above-mentioned catalytic system, I could not find the innovation and importance of this one and is only an addition to the field. What is more, the authors said, EC provide valuable insights for the design of electrocatalysts toward small organic molecule catalysis ED;. Questions like what happens to the other types of small molecule nitriles must be addressed.*

Response: We thank the referee for the valuable comments, and we are glad to clarify this issue. For the two papers (Nat. Commun. 2021, 12, 1949, Chem Catal. 2021, 1, 393-406) mentioned by the referee, metallic copper was used for converting nitriles to primary amines, and the energetic and Faradic efficiencies at industrial level current densities is not good enough (55.7 % at 1000 mA/cm²), although good performance has been achieved at low current

density region. Moreover, the catalytic mechanism of the mentioned papers is built based on metallic copper. As comparison, our work uncovers that lattice oxygen in copper plays vital roles in modulating the key intermediate adsorption and boosting the catalytic conversion from nitriles to primary amines. By employing various advanced static and dynamic characterization techniques, we reveal that the high abundance of free electrons on the metallic copper leads to strong electronic coupling with amines and therefore hinders its catalytic kinetics, while electron localization effect created by lattice oxygen can restrict the flow of surface electrons in copper surface and further reduce the binding affinity to amines. Meanwhile, the fabricated membrane electrode assembly (MEA) demonstrates the potential application of OD-Cu NWs for nitrile reduction to primary amines at industry-scale current (~80% at 1.6 A).

Evaluating the importance or novelty of a work should rely on the solved scientific or technical questions. In comparison with the mentioned papers which used copper as catalyst for hydrogenation of nitriles, our work targets a different scientific question. We used lattice oxygen to solve the faced adsorption kinetics of metallic coppers and achieved better catalytic performance. Since the developed strategy is conceptually new for nitrile reduction, we reasonably believe the novelty of our work is suitable for Nature communications.

Following the suggestions of the referee, we furthermore expanded the developed strategy to other nitrile hydrogenations, including Cyclopropanecarbonitrile (CPN), 3-Hydroxypropionitrile (3-HPN), Butyronitrile (BN), and Pentanenitrile (PN), as shown in **Fig. R1 (detail in Supplementary Figs. 20-23)**. For all the studied small molecular nitriles, the conversions to amines on OD-Cu NWs are always better than on Cu NWs, with both higher current densities and Faradic efficiencies, which further demonstrates residual oxygen in OD-Cu NWs can efficiently manipulate the catalytic reduction of nitriles to primary amines. For the hydrogenation of butyronitrile (BN) and pentanenitrile (PN), the lower FEs of 34.5% and 22.6% are mainly due to their lower solubility.

In summary, this work provides a new approach to further improve the performance of the hydrogenation of nitriles, by employing lattice oxygen-mediated electron tuning engineering. We have updated the data in the revised manuscript as **Supplementary Fig. 19**, and also given a brief discussion on it.

Fig R1 | Electrochemical hydrogenation of nitriles (Cyclopropanecarbonitrile (CPN), 3-Hydroxypropionitrile (3-HPN), Butyronitrile (BN), and Pentanenitrile (PN)).

2. Much of the modeling and interpretation of the catalytic performance could be overturned if the coordination number (CN) of Cu-Cu, rather than the residual lattice oxygen, is proven to cause the differences in electrochemical performance between OD-Cu NWs and Cu NWs. The lattice oxygen, which is bonded with Cu as depicted in the inner graphic of Fig. 3b, is not detected either in OD-Cu NWs or Cu NWs (Fig. 3e). The significant difference between them is the CN of Cu-Cu (9.1 vs 10.3). The author should better explain about this point.

Response: We are grateful to the referee for the insightful comments. First, to prove the existence of lattice oxygen, Raman spectroscopy, Synchrotron radiation photoelectron spectroscopy (SRPES) and X-ray absorption spectroscopy (XAS) were employed. As displayed by the Raman spectra of OD-Cu NWs in Fig. 3a, the signal at 389.9 cm⁻¹ is attributed to Cu-O vibrational mode, indicating the presence of O inside the lattice. In addition, based on the SRPES spectra at Cu 3p, the valence state of near-surface Cu atoms in OD-Cu NWs is slightly higher than the inner atoms, suggesting the incorporation of O. Furthermore, the existence of O in OD-Cu NWs is verified by O-K edge XAS (Fig. 3b), where the peak at 533.28 eV belongs to Cu₄-O type lattice oxygen. For Cu NWs, on the contrary, neither Cu-O bond is

detected based on Raman and SRPES spectra, nor did significant O exist based on the O-K edge XAS.

To exclude the effects of coordination numbers on the catalytic performance, we used both experimental and theoretical methods. Experimentally, considering nanoparticles with larger size are generally supposed to possess higher coordination number (CN) and the smaller one possesses lower CN, we have conducted a comparative analysis of two copper nanoparticles with distinct particle sizes (Cu NP (200 nm) and Cu NP (100 nm)). TEM and XRD analyses show the average particle diameters of Cu NP (100 nm) and Cu NP (200 nm) are ~100 nm and ~200 nm, respectively (**Figs. R2a-c**). The catalytic performance of both nanoparticles for the acetonitrile reduction with the same catalyst loading are assessed. Over the investigated potential range, compared with Cu NP (200 nm), Cu NP (100 nm) delivers lower current density, less attractive Faradaic efficiency (FE) and unsatisfying partial current density for acetonitrile reduction to ethylamine, particularly in the high potential region (<-0.55 V vs RHE) (**Fig. R2d-g**), demonstrating limited catalytic performance on Cu NP (100 nm). Thus, it seems that Cu NPs with lower coordination number could even obstacle the acetonitrile hydrogenation, suggesting that the changed coordination number (from 10.1 of Cu NWs to 9.3 of OD-Cu NWs) might not be critical factor for the highly efficient acetonitrile hydrogenation on OD-Cu NWs. Therefore, the difference of the catalytic activity is supposed to originate from the existence of lattice oxygen residues.

Furthermore, we have also theoretically investigated the impact of unsaturated surface coordination (100, 211, 110) by constructing high-index facets and unsaturated bulk coordination (111-d) by creating Cu vacancies beneath the surface for acetonitrile reduction (**Fig. R3a**). Our results show that both unsaturated surface coordination and unsaturated bulk coordination exhibit stronger adsorption of amine intermediates (CH_3CN^* , CH_3CHN^* , CH_3CNH^* , CH_3CHNH^* , $\text{CH}_3\text{CH}_2\text{N}^*$, $\text{CH}_3\text{CNH}_2^*$, $\text{CH}_3\text{CH}_2\text{NH}^*$, $\text{CH}_3\text{CH}_2\text{NH}_2^*$ and H_2O^*) than the O-bearing B-OD-Cu surface, particularly for ethylamine adsorption, which is detrimental to the desorption of $\text{CH}_3\text{CH}_2\text{NH}_2$ from the catalyst surface and thus hinders the catalytic kinetics (**Figs. R3b**).

In conclusion, combining the experimental and computational results, the changes of coordination number might not be the determining factor for efficient conversion of AN to generate EA. Instead, the oxygen residues existing in the form of $\text{Cu}_4\text{-O}$ could be the key factor for improving the catalytic kinetics of nitrile hydrogenation.

We have incorporated the suggestions made by the reviewer into the revised manuscript and have added corresponding discussions. Additionally, **Figs. R2 and R3** have been included in the Supporting Information as **Supplementary Fig. 39** and **Supplementary Fig. 40**.

Fig. R2| Characterizations and catalytic performances of Cu NP (100 nm) Cu NP (200 nm). **a,b**, TEM images of Cu NP (100 nm) (a) and Cu NP (200 nm) (b). **c**, XRD patterns of Cu NP (100 nm) and Cu NP (200 nm). **d**, Linear sweep voltammetry (LSV) plots of Cu NP (100 nm) and Cu NP (200 nm) in argon-saturated 1 M KOH aqueous solutions with and without the addition of acetonitrile (AN) (8 wt%). Scan rate: 10 mV s⁻¹. **e,f**, Faradic efficiencies (FE) of various reduction products at different potentials on (e) Cu NP (100 nm) and (f) Cu NP (200 nm). **g**, Partial current densities of ethylamine at different potentials on Cu NP (100 nm) and Cu NP (200 nm).

Fig. R3 | Impact of oxygen residues and coordination configurations on Cu catalyst performance in acetonitrile reduction for ethylamine production. a, Cu catalysts with various surface and bulk coordination configurations, including Cu(100), Cu(110), Cu(211), and Cu(111)-d. **b,** Comparison of the adsorption energies of critical reaction intermediates involved in the acetonitrile reduction reaction to form ethylamine among the Cu catalysts with different coordination configurations and oxygen residues.

3. The results of Supplementary Table 1 and 2 are in conflict. OD-Cu NWs is well fitted to 27.8% Cu_2O with Cu_4-O coordination and 72.2% Cu foil in Supplementary Table 1, but the fitting results in Supplementary Table 2 do not support the exit of Cu-O coordination of OD-Cu NWs.

Response: We thank the referee for the insightful comment and we are glad to clarify this issue. **Supplementary Table 1** and **Table 2** uncover the structure from two different points and the results might be different yet not contradictory. In supplementary **Table 1**, the linear combination fitting (LCF) of OD-Cu NWs with introducing Cu_4-O coordination is rationalized by the presence of Cu_2O -like Cu_4-O interaction, as verified by Raman spectroscopy, O-K edge

XAS and SRPES. Based on the LCF fitting result, only 12.2 at% oxygen exists in the system with 27.8% Cu₄O coordination and 72.2% metallic Cu configuration, which is hard to be detected by EXAFS (**Fig. R4 a**). One step further, the k-space data derived from the LCF-fitting of E-space is Fourier transformed into R-space, as presented in **Fig. R4 b, c**. Although the definite 12.2 at% O atoms exist in the LCF-fitting data contains, no significant Cu-O bond information is observed in its R-space, which is in accordance with the OD-Cu NWs. Thus, although 27.8% Cu₄O coordination and 72.2% metallic Cu configuration exist in the LCF-fitting results, the content of O is too low to be detected by EXAFS.

Fig. R4 has been added in Support Information as **Supplementary Fig. 11** and corresponding discussion has been added accordingly in the revised manuscript.

Fig. R4 | XANES and EXAFS analysis of OD-Cu NWs. a, Cu K-edge linear component fitting (LCF) XANES spectrum. **b, c** k^3 -weighted EXAFS (b), and corresponding FT-EXAFS (c) from the LCF results.

4. The EXAFS fitting results in Supplementary Table 2 show the R of Cu-Cu is 2.54; for both OD-Cu NWs and Cu NWs. However, the XRD, HRTEM, and HAADF-STEM results in Fig. 2 indicate an obvious lattice expansion in OD-Cu NWs as compared to Cu NWs. These results seem also contradictory.

Response: We appreciate the valuable comments and we are pleased to clarify this issue. The R value of EXAFS only provides average structural information of the bulk and is less sensitive to the local structure change. In addition, due to the limited accuracy of EXAFS, small changes might not be easily detected. In OD-Cu NWs, the moderate lattice residual oxygen dose not significantly change the lattice structure and the amount of O residue is low, which could explain the nearly unchanged R value of OD-Cu NWs compared with Cu NWs. Similarly, X-ray diffraction (XRD) also provides bulk information about the crystal structure and periodicity of materials. Based on the experimental interplanar spacing calculated by the Bragg equation, the (111) interplanar distances of Cu NWs and OD-Cu NWs are 0.2087 nm and 0.209 nm,

respectively. Thus, although small lattice expansion is observed in XRD, such small value difference makes it almost impossible to see the R value change in EXAFS spectra. However, unlike EXAFS or XRD, the HRTEM images are more sensitive to local structure and could provide more apparent evidence on lattice expansion. Meanwhile, to obtain structural information of the surface where catalytic reaction happens, the edges of the OD-Cu NWs are studied in the HRTEM tests. Compared with Cu NWs, lattice expansion occurs on the surface of OD-Cu NWs. Furthermore, **Fig. R5** displays HRTEM image of OD-Cu NWs at larger scale, which both contains edge and bulk information. Compared with Cu NWs, there is only negligible change in interplanar spacing in bulk-phase OD-Cu NWs (0.208 nm), which is consistent with XAFS results, while more apparent changes happen at the edge (0.214 nm).

Overall, it is necessary to employ multiple techniques and analytical methods in order to obtain a comprehensive understanding of the material properties. A corresponding discussion has been added accordingly in the revised manuscript.

Fig. R5 | The HRTEM image of OD-Cu NWs.

5. *The comparison should be more accurate if the O K-edge NEXAFS of Cu NWs is provided in Fig. 3b.*

Response: We thank the referee for the valuable comment. Based on the suggestions of the referee, the O K-edge NEXAFS of Cu NWs has been provided in **Fig. 3b**, and the corresponding discussion has been included in the relevant section of the manuscript. In order to avoid the surface oxidation by air exposition, all the samples have been carefully stored and transferred by a sealed cell at the same condition. Due to the ultralow amount of oxygen, experimental results show a weak O K-edge signal on Cu NWs, obviously different from OD-

Cu NWs that possess considerable Cu₄O-like O atoms. Thus, the evidence on the existence of O residues in OD-Cu NWs would be strengthened.

6. The characterizations of the catalysts after electrocatalytic hydrogenation of nitriles should be studied to determine if they have changed during the course.

Response: We sincerely appreciate the reviewer's insightful suggestion. The structure of OD-Cu NWs catalysts after electrocatalytic tests has been investigated. Specifically, based on scanning electron microscopy (SEM), transmission electron microscopy (TEM), and X-ray diffraction (XRD) (**Fig. R6 a, b**), the OD-Cu NWs maintained their initial nanowire morphology and fcc-Cu phase after the test. Furthermore, as displayed by the Raman spectra in **Fig. R6c**, a weak band at around 389 cm⁻¹ is still present in the tested OD-Cu NWs, indicating the presence of residual lattice oxygen in the tested OD-Cu NWs sample. In addition, the core-level XPS and Cu L3M45M45 Auger spectra of the tested OD-Cu NWs were also measured to study the valence changes. Essentially, the tested OD-Cu NWs exhibited clear lattice metal-O bond (530.6 eV, **Fig. R6 d**) and Cu^{δ+} (916.5 eV, **Fig. R6 f**) signals, indicating the presence of lattice oxygen residues in the OD-Cu NWs. We have added the corresponding description in the revised manuscript and supporting information. Meanwhile, **Fig. R6** has been added in Support Information as **Supplementary Fig. 27**.

Fig. R6 | a, SEM and TEM images of OD-Cu NWs before and after the test. b, XRD patterns of OD-Cu NWs before and after the test. c, Raman spectra of OD-Cu NWs before and after the

test. **d-f**, XPS spectra of O 1s (**d**), Cu 2p (**e**) and corresponding Auger spectra of OD-Cu NWs (**f**) before and after the test.

7. *The OD-Cu NWs with lattice oxygen are prepared at -0.37 V vs. RHE, which is the reduction potential of Cu₂O to Cu. However, the applied potential for the hydrogenation of nitriles is more negative than -0.37 V (Fig. 4b). And, what are the cathodic potentials when conducting the hydrogenation reaction of nitriles at the current density of more than 1.5 A in Fig. 4d? At such negative potentials or high current densities, can the lattice oxygen in OD-Cu stably exist? Or in fact, the real catalyst for the hydrogenation of nitriles is not the same as the OD-Cu NWs discussed in this work.*

Response: Thank you for your kind suggestion, and we are glad to clarify this issue. Firstly, **Fig. 4d** shows the full electrolyzer voltage measured by the two-electrode AEM-MEA testing system, which is 2.54 V at the current of 1.5 A. Unfortunately, considering the voltage of the electrolyzer is jointly contributed by the overpotentials of the electrodes, the Ohmic drop and the mass transport loss, it is hard to solely determine the exact potential of the cathode at such large current.

However, to address the reviewer's concern regarding the stability of the lattice oxygen under the cathodic reaction, we've conducted in-situ Raman spectroscopy to monitor the changes of the catalyst throughout the reaction process (**Fig. R7**). The presence of Cu-O vibrational mode is maintained throughout the entire test duration, even at high cathodic potentials of up to -1.05 V vs RHE (without *iR* compensation), indicating the existence of lattice oxygen residues throughout the reaction process. Thus, at the potential even more negative than the reduction potential of Cu₂O to Cu, the lattice oxygen residues in the OD-Cu NWs are stable throughout the acetonitrile hydrogenation reaction, which could rationalize the identification of OD-Cu NWs as the real catalyst during the acetonitrile reduction.

Fig. R7 has been added in Support Information as **Supplementary Fig. 30** and corresponding discussion has been added in the revised manuscript.

Fig. R7 | The in-situ Raman spectra of OD-Cu NWs at various potentials.

8. *The adsorption modes, such as adsorption via only the N atom or both the C and N atoms, of various molecules should be optimized and illustrated because different adsorption models lead to different shifts in FTIR and Raman spectra (J. Phys. Chem. C 2019, 123, 2300-2313). The simulated results should be compared with the experiment data to exactly determine the adsorption of the reaction species, hence a more accurate reaction mechanism.*

Response: We thank the referee for the valuable comment. In response to the reviewers' comments, the infrared spectra of all intermediate species involved in the electroreduction of acetonitrile to ethylamine in their most stable adsorption configurations on Cu(111) and B-OD-Cu(111) surfaces are further investigated using the symmetry-based density functional perturbation theory (DFPT) method (**Fig. R8 b,d**). The results are compared with the experimental in-situ SR-FTIR spectra (**Figs. R8 a,c**). Among the molecules considered in the calculation, the computational IR spectrum of ethylamine adsorbed on the Cu (111) surface (EA*) matches well with the in-situ SR-FTIR results obtained on Cu NWs throughout the potential region during the test. It can be inferred that the desorption of EA* is the potential determining step (PDS) on Cu(111) surface in the reaction due to its strong adsorption. In contrast, based on the calculated IR spectra on B-OD-Cu(111) surface, the signal of CH₃CHNH* is consistent with the experimental SR-FTIR results from OD-Cu NWs within the testing potential range (**Figs. R8 c,d**). Thus, on B-OD-Cu(111), the desorption of EA could be more favorable than on Cu(111), and the PDS is shifted to the proton-coupled electron transfer electrochemical process of CH₃CHNH* to CH₃CH₂NH*. The detailed analysis of the calculated IR spectra and experimental SR-FTIR spectra is included in the revised manuscript. **Fig. R8** has been added in Support Information as **Supplementary Fig. 37**.

Fig. R8 | a-d, In-situ SR-FTIR spectra of Cu NWs (a) and OD-Cu NWs (c) at different potentials and IR spectra of intermediates involved in acetonitrile electroreduction on Cu(111) (b) and B-OD-Cu (111) (d).

9. The $\nu(\text{NH}_2)$ vibration cannot be distinguished because it is overlapped with the $\nu(\text{OH})$ vibration in Supplementary Fig. 20a.

Response: We thank the referee for the valuable comment. Distinguishing $\nu(\text{NH}_2)$ from $\nu(\text{OH})$ in vibrational spectroscopy remains a challenge because the two vibrations overlap together. Besides the $\nu(\text{NH}_2)$ vibration, other vibrational modes attributed to $\nu(\text{N-H})$ and $\nu(\text{C-N})$ of the key intermediates are detected at $\sim 1100 \text{ cm}^{-1}$ and $\sim 1400 \text{ cm}^{-1}$, can be used for proving the existence of NH_2 and EA^* (**Fig. R9 a, b**). As shown in **Fig. R9 a**, we can clearly observe that the strong $\nu(\text{N-H})$ and $\nu(\text{C-N})$ vibrational signal of Cu NWs gradually increases as the applied potential further increases from -0.17 V to -1.07 V and finally remains almost constant, verifying that the strong adsorption of EA^* on metallic Cu leads to a large coverage of NH -containing species and unfavorable desorption of EA^* . Conversely, no ethylamine adsorption peak was detected on OD-Cu NWs. The trends of the $\nu(\text{N-H})$ at $\sim 1100 \text{ cm}^{-1}$ and $\nu(\text{C-N})$ at

$\sim 1400\text{ cm}^{-1}$ are well consistent with that of $\nu(\text{NH}_2)$ vibration at $\sim 3000\text{ cm}^{-1}$. **Fig. R9** has been added in Support Information as **Supplementary Fig. 28** and corresponding discussion has been added in the revised manuscript.

To distinguish the contribution of $\nu(\text{NH}_2)$ vibration and the $\nu(\text{OH})$ vibration, Gaussian deconvolution method has been used to fit the IR spectrum¹. According to the experimental data and theoretical simulations of EA, the $\nu(\text{NH}_2)$ exists at approximately 3000 cm^{-1} (**Fig. R9 c**). Furthermore, the experimental signal within $2800\text{-}3750\text{ cm}^{-1}$ are quantitatively fitted in the region of $2800\text{-}3750\text{ cm}^{-1}$, where the presence of NH_2 is observed on Cu NWs but not exists on OD-Cu NWs (**Fig. R9 d**).

Fig. R9 | **a,b** In-situ SR-FTIR spectra of Cu NWs (**a**) and OD-Cu NWs (**b**) at different potentials. **c,d**, Experimental and theoretical simulations of infrared spectra of ethylamine solutions (**c**), and peak fitting of Cu NWs and OD-Cu NWs in the $2800\text{-}3750\text{ cm}^{-1}$ SR-FTIR spectral region at -1.07 V vs. RHE (without iR compensation) (**d**).

Reference

1. Branca, C. *et al.* Role of the OH and NH vibrational groups in polysaccharide-

nanocomposite interactions: A FTIR-ATR study on chitosan and chitosan/clay films. *Polymer* **99**, 614-622 (2016).

10. *Moreover, there are some mistakes in the main text, such as the $\acute{E}C$; desorption $\acute{E}D$; in line 76 should be adsorption, there is an extra $\acute{E}C$;b $\acute{E}D$; in line 231, and $\acute{E}C$;j EA $\acute{E}D$; in line 243 should be italicized.*

Response: We really thank the reviewer for the carefully review and pointing out these mistakes. We have corrected these mistakes and meanwhile we have also double-checked the manuscript to avoid such mistakes.

11. *Does this electrocatalytic hydrogenation method apply to other nitriles? I'm just giving a suggestion.*

Response: We greatly appreciate the valuable feedback provided by the referee and have taken their suggestion to investigate the impact of residual oxygen in OD-Cu NWs on other nitrile hydrogenations into consideration. Specifically, we examined the effects of residual oxygen on the hydrogenation of cyclopropanecarbonitrile (CPN), 3-Hydroxypropionitrile (3-HPN), butyronitrile (BN), and pentanenitrile (PN), and have summarized the results in **Fig. R1**. The results show that residual oxygen in OD-Cu NWs can effectively catalyze the highly selective production of amines with $C\equiv N$ bond in these reactions. The lower FEs of 34.5% and 22.6% during the hydrogenation of BN and PN originate from their lower solubility.

We have incorporated the suggestions made by the reviewer into the revised manuscript and have added corresponding discussions. Additionally, **Fig. R1** have been included in the Supporting Information as **Supplementary Fig. 19**. Also all nitrile test details have been added in Supporting Information as **Supplementary Figs. 20-23**.

Reviewer #2 (Remarks to the Author): *In this manuscript, wei and co-workers reveal that residual lattice oxygen in oxide-derived Cu nanowires (OD-Cu NWs) plays vital roles in boosting the acetonitrile electroreduction efficiency. A series of in situ characterization techniques were applied to understand the role of oxygen residual in the catalytic process. Theoretical calculations suggest that oxygen residues act as electron acceptors to confine the free electron flow on the Cu surface, consequently improving the desorption of ethylamine. Overall, this work is interesting and well presented, and the understanding on the effect of oxygen residual is also systematical and comprehensive by using various techniques. Before acceptance, necessary revisions are required:*

Response: We sincerely thank the referee for carefully reviewing our manuscript and recognizing the understanding on the effect of oxygen residual is systematical and comprehensive by using various techniques in this work. We also appreciate the valuable comments which certainly help to improve our manuscript. We have prepared a point-by-point response to address the raised concerns, which are presented below.

1. *The authors claim that oxide-derived copper has better performance due to the residue oxygen in copper lattice. However, Cu is an active metal especially unsaturated Cu after electrochemical reduction which could be easily oxidized. In situ characterizations should be added to exclude the residual oxygen which is not from the external environment.*

Response: We thank the referee for the insightful comment. In response to the reviewer's concern about the possibility of residual oxygen from the external environment, we have used in-situ Raman spectroscopy to monitor the Cu structure changes during the reaction process (**Fig. R1**). Our results show that the Cu-O vibrational mode persists throughout the potential test range, even at cathode potentials up to -1.05 V vs. RHE (without *iR* compensation). In order to avoid the surface oxidation by air exposition, all the samples have been carefully stored and transferred by a sealed cell at the same condition. For example, the O 1s XPS, Cu 2p XPS, and Cu auger XPS spectra demonstrate the presence of lattice oxygen and Cu^{δ+} even after testing, further confirming the existence of lattice oxygen in OD-Cu NWs (**Figs. R2 a-c**). Overall, these results further prove that the lattice oxygen residues observed in the OD-Cu NWs are stable during the reaction and the detected O signals are not solely from the external environment.

We have incorporated the suggestions made by the reviewer into the revised manuscript and have added corresponding discussions, highlighting the changes made. Additionally, **Fig.**

R1 and Fig. R2 have been included in the Supporting Information as **Supplementary Fig. 30** and **Supplementary Fig. 27**.

Fig. R1 | The in-situ Raman spectra of OD-Cu NWs at different potentials.

Fig. R2 | a-c, XPS spectra of O 1s (a), Cu 2p (b) and corresponding Cu L3M45M45 Auger spectra of OD-Cu NWs (c) before and after the test.

2. In the DFT calculation, two models are built to simulate the OD-NW. The authors claim that the second model is chosen according to the experimental characterization results but did not explain the reason. And why only underlayer oxygen is considered and surface oxygen is not considered?

Response: We appreciate the valuable feedback from the referee. First of all, the selection of B-OD-Cu(111)-O_{sub} was based on the static structural characterization (Raman, SRPES and XAS) and AIMD simulations. Specifically, based on the experimental results of OD-Cu NWs, about 12.2 at% residual oxygen exists in the form of Cu₄-O structure and creates Cu^{δ+}. Subsequently, OD-Cu catalyst models with different oxygen content were constructed based on these experimental results, and the structures were evaluated by AIMD studies. The bond length and coordination number of B-OD-Cu(111)-O_{sub} match well with the static experimental

characterizations, while electronic structure analysis also revealed the presence of $\text{Cu}^{\delta+}$ inside B-OD-Cu(111)- O_{sub} , similar to the experimental results. Therefore, B-OD-Cu(111)- O_{sub} was selected as the model for DFT calculations.

To answer the question raised by the referee "*why only underlayer oxygen is considered and surface oxygen is not considered?*", we performed DFT calculations to investigate the effect of surface oxygen-containing species on the electrochemical reduction of acetonitrile to ethylamine. **Fig. R3 a,b** shows the configurations of OH and O adsorbed on the Cu (111) surface ($\text{O}_{\text{ads}}@\text{Cu}(111)$ and $\text{OH}_{\text{ads}}@\text{Cu}(111)$), and the adsorption energy of all molecular intermediates involved in the reaction. When oxygen-containing species are present on the Cu (111) surface, the adsorption strengths of amine intermediates are slightly enhanced compared with pure Cu(111) surface and much higher than on B-OD-Cu(111) surface. Furthermore, the effect of surface-adsorbed oxygen-containing species on the reaction mechanism of the hydrogenation of acetonitrile to ethylamine is investigated. Similar to Cu(111), due to the strong adsorption of $\text{CH}_3\text{CH}_2\text{NH}_2^*$, the desorption of $\text{CH}_3\text{CH}_2\text{NH}_2^*$ is the potential-dependent step (PDS) on $\text{O}_{\text{ads}}@\text{Cu}(111)$ and $\text{OH}_{\text{ads}}@\text{Cu}(111)$ surfaces. In contrast, due to the weakened $\text{CH}_3\text{CH}_2\text{NH}_2^*$ adsorption, the desorption of $\text{CH}_3\text{CH}_2\text{NH}_2^*$ is favored, and the PDS is switched from the chemical process of $\text{CH}_3\text{CH}_2\text{NH}_2^*$ desorption into the proton-coupled electron transfer electrochemical processes of CH_3CHNH^* to $\text{CH}_3\text{CH}_2\text{NH}^*$ (**Fig. R3 c**).

In conclusion, the local surface electron distribution caused by lattice oxygen residues rather than surface O-containing species is key to enhancing the electroreduction of acetonitrile to ethylamine on the Cu metal surface.

We have added corresponding discussions, highlighting the changes made. Additionally, **Fig. R3** have been included in the Supporting Information as **Supplementary Fig. 41**.

Fig. R3 | **a**, Configuration of Cu catalysts adsorbing different oxygen-containing species (OH_{ads}@Cu(111) and O_{ads}@Cu(111)). **b**, The adsorption energies of the key intermediates involved in acetonitrile reduction reaction on the Cu catalysts with different oxygen-containing species. **c**, Potential energy diagram of the optimal formation route for the generation of EA by AN electroreduction on the OH_{ads}@Cu(111) and O_{ads}@Cu(111) surfaces at 0 V (vs RHE).

3. The O K-edge NEXAFS of Cu NWs should be provided for comparison. In addition, detailed valence states of Cu should be calculated in order to better understand the effect of O residues on the electronic structure of Cu.

Response: Thanks for your kind suggestion. Following their suggestions, we have included the O K-edge NEXAFS spectrum of Cu NWs in **Fig. 3b** of the revised manuscript, and calculated the detailed Cu oxidation state based on XAFS spectra (**Supplementary Fig. 9**) in the revised Supplemental Information. Relevant discussions have been included in the manuscript and have been emphasized. In order to avoid the surface oxidation by air exposition, all the samples have been carefully stored and transferred by a sealed cell at the same condition. Due to ultralow amount of oxygen, the experimental results show a weak O K-edge signal on Cu NWs, obviously different from OD-Cu NWs that possesses considerable Cu₄-O like O atoms. The

detailed valence states of the Cu catalysts show that the valence state of Cu in OD-Cu NWs is 0.46 (**Fig. R4**). Thus, the evidence on the existence of O residues in OD-Cu NWs would be strengthened.

Fig. R4 | The detailed valence states of Cu catalysts.

4. The authors should specify which kind of AEM is used in this study and how is the stability of AEM during the electrolysis.

Response: Thanks for your kind suggestion. Our study used an AEM made of NPPO-2QA-x polymer, which is a modified PPO polymer backbone polymerization with 1,3-bis(trimethylammonium-bromide-methyl)-5-(prop-2-ynyloxy) benzene (TABB) and azide modifications.¹ The AEM has a thickness of 50 μm . During electrolysis of acetonitrile at a total current of 1.6 A, the AEM maintained its original performance for 16 hours in a 1 M KOH solution, indicating good stability. In addition, according to the reviewer's comments, we have added the membrane information in method part.

Reference:

1 Song, W. *et al.* Hydrogen bonding assisted OH⁻ transport under low humidity for rapid start-up in AEMFCs. *J. Membr. Sci.* **647**, 120303 (2022).

5. Necessary information should be clearly presented. For example, the attribution of each peak in Supplementary Fig. 21.

Response: We thank the referee for the valuable comment. Based on the reviewer's comments, we have added the attribution of each peak to **Supplementary Fig. 29**. **Fig. R5** shows the added **Supplementary Fig. 29**.

Fig. R5 | a,b, 3D and 2D plots of in situ Raman spectra of OD-Cu NWs at different potentials.

Reviewer #3 (Remarks to the Author): *The manuscript entitled “Lattice oxygen mediated electron tuning promotes electrochemical hydrogenation of acetonitrile on copper catalysts” by Wei, et al. reported a study of the electrocatalytic nitriles hydrogenation primary amines on Cu-based catalysts. In particular, the authors attributed the observed high acetonitrile electroreduction efficiency to the residual lattice oxygen in oxide-derived Cu nanowires (OD-Cu NWs). They identified that the “oxygen residues, in the form of Cu₄-O configuration, act as electron acceptors to confine the free electron flow on the Cu surface, consequently improving the kinetics of nitriles hydrogenation catalysis.” A key claim of the manuscript is that the “oxygen residues” promote “electron confining capability of copper atoms” and weakens the amine binding strength. On the other hand, desorption of the amine is a nonelectrochemical step and the characterizations do not provide convincing support of the promotion effect. In summary, although this manuscript reported some improvements over the previous publications, the overall progress is incremental and does not warrant its publication in Nature Communication.*

Response: We thank the referee for recognizing this manuscript with improvements over the previous publications, and we also appreciate their valuable comments, which are certainly helpful for improving our manuscript. We have prepared a point-by-point response to address the issues raised, as detailed below.

To support the promotion effect of the lattice oxygen, we used in-situ SR-FTIR to investigate the electroreduction of acetonitrile to generate ethylamine on Cu NWs and OD-Cu

NWs (**Figs. R1 a, b**). As the cathodic potential increases during the acetonitrile reduction reaction, the strong $\nu(\text{NH}_2)$ vibration signal of Cu NWs gradually increases and then remains almost unchanged, verifying that the strong adsorption of EA* on metal Cu could lead to large-scale coverage of NH_2 -containing species and make it unfavorable for the desorption of EA*. Impressively, upon the introduction of lattice oxygen residues, there appears no visible -NH_2 band vibration on OD-Cu NWs as the applied potential increases, indicating that the binding strength of -NH_2 on OD-Cu NWs is moderate, and the desorption of -NH_2 -containing species is not significantly hindered.

Furthermore, we further theoretically calculated the infrared spectra (IR) of all intermediate species involved in the electroreduction of acetonitrile to ethylamine in their most stable adsorption configurations using the symmetry-based density functional perturbation theory (DFPT) method and correlated them with the experimental in-situ SR-FTIR results. **Figs. R1c** shows the IR of all involved intermediate species adsorbed on Cu (111) surface. The calculated IR spectra of ethylamine (EA) adsorbed on the Cu(111) (Cu NWs) surface is consistent with the in-situ SR-FTIR spectra of Cu NWs at all potentials, indicating that the adsorption of EA is the potential-determining step (PDS) for the electroreduction of acetonitrile to EA. Nevertheless, according to the IR study of all intermediates involved in the electroreduction of acetonitrile to ethylamine on B-OD-Cu(111) (OD-Cu NWs), the spectra of CH_3CHNH^* is close to in-situ SR-FTIR spectra obtained on OD-Cu NWs, and thus the transformation of CH_3CHNH^* is considered as the PDS for the electroreduction of acetonitrile to ethylamine on OD-Cu NWs. Overall, it is evident that the desorption of EA is the potential determining step (PDS) in the reaction due to its strong adsorption on Cu NWs. In contrast, on OD-Cu NWs (B-OD-Cu (111)) (**Fig. R1 d**), where the adsorption of EA is weakened, the desorption of EA is favored, and the PDS is shifted from the chemical process of EA desorption to the proton-coupled electron transfer electrochemical process of CH_3CHNH^* to $\text{CH}_3\text{CH}_2\text{NH}^*$.

Evaluating the importance or novelty of a work should rely on the solved scientific or technical questions. In comparison with the previous papers which used only metallic copper for catalytically converting nitriles to primary amines, our work targets a different scientific question. We used lattice oxygen to solve the faced adsorption kinetics of metallic coppers, and achieved better catalytic performance. By employing various advanced static and dynamic characterization techniques, we reveal that the high abundance of free electrons on the metallic copper leads to strong electronic coupling with amines and therefore hinders its catalytic

kinetics, while electron localization effect created by lattice oxygen can restrict the flow of surface electrons on copper surface and further reduce the binding affinity to amines. Since the developed strategy is conceptually new for nitrile reduction, we reasonably believe the novelty of our work is suitable for Nature communications.

We have added corresponding discussions, highlighting the changes made. Additionally, **Fig. R1** have been included in the Supporting Information as **Supplementary Fig. 37**.

Fig. R1 | **a-d** In-situ SR-FTIR spectra of **(a)** Cu NWs and **(c)** OD-Cu NWs at different potentials, as well as IR spectra of all intermediate species involved in the electroreduction of acetonitrile to ethylamine in their most stable adsorption configuration of **(b)** Cu(111) and **(d)** B-OD-Cu (111).

Specific comments:

1. *The manuscript quoted a total current density of 50 mA cm⁻² but failed to point out the partial EA current density is ~0.85 A cm⁻² for ref. 17.*

Response: Thanks for your kind suggestion. We have added the missed information in the revised manuscript.

2. The manuscript did not report hydrogen production in the experimental results although proton reduction was included in the mechanistic study. What is the EA selectivity?

Response: We thank the referee for the valuable comment. Following their suggestions, the FE efficiencies of H₂ have been supplemented in the revised Supplemental Information, and the corresponding test details are given in the methods section. Relevant discussions have been included in the manuscript and have been emphasized. Faraday efficiencies and partial current density analyses of the various reduction products at different potentials on OD-Cu NWs and Cu NWs indicate that ethylamine is the dominant product on OD-Cu NWs and Cu NWs (**Fig. R2**). The FE of OD-Cu NWs remained above 90% throughout the potential region studied, while Cu NWs showed significantly lower FE values, especially in the high potential region due to competing reactions of HER. In addition, exploration of the selectivity of ethylamine for all amines showed that the selectivity of ethylamine was maintained at ~99% over the range of potentials tested. These results suggest that the residual lattice oxygen in the OD-Cu NWs plays a key role in the catalytic activation of acetonitrile to form ethylamine.

We have added corresponding discussions, highlighting the changes made. Additionally, **Fig. R2** have been included in the Supporting Information as **Supplementary Fig. 18**.

Fig. R2 | Catalytic performances of OD-Cu NWs and Cu NWs samples in the acetonitrile reduction reaction. a-d, Faraday efficiencies and partial current densities of various reduction products at different potentials on OD-Cu NWs (**a,c**) and Cu NWs (**b,d**)

3. *The authors determined that the subsurface oxygen in the form of “B-OD-Cu” is the active configuration for EA formation. Is this model consistent with the oxygen concentration in the OD-Cu NWs?*

Response: We appreciate the constructive comments. The oxygen concentration in B-OD-Cu model is consistent with that in OD-Cu NWs. Our experimental results including static structure and electronic state characterization (Raman, SRPES and XAS) demonstrate the existence of residual oxygen with Cu₄-O configuration in electrochemically reduced OD-Cu NWs, with an oxygen content of about 12.2 at% and an average Cu-Cu coordination number of 9.1. Subsequently, models of OD-Cu with different oxygen contents were constructed based on the experimental results. After AIMD simulations, the bond length and coordination structure of B-OD-Cu(111)-O_{sub} with 11 at% oxygen match well with the static experimental characterization (XAFS), which rationalizes the model we constructed.

The authors quote a CN of 4.4 for B-OD-Cu model. Is this local coordination number for the O atom buried in the subsurface hollow site? How does this compare with the averaged O-Cu coordination number from XAFS?

Response: We thank the referee for the valuable suggestion, and we are glad to clarify this issue. First, the CN of the B-OD-Cu model is 4.4, which represents the local coordination number of buried subsurface O atoms. Due to the unavailability of O extended X-ray absorption fine structure data, it is challenging to accurately determine the O-Cu coordination number. Notably, we qualitatively revealed the presence of residual oxygen with Cu₄-O type configuration in the OD-Cu NWs by using X-ray absorption spectroscopy (XAS) linear combination fitting (LCF), O K-edge NEXAFS. In addition, the average coordination number of Cu-O was analyzed using radial distribution function (RDF) analysis (**Fig. R3**), and the result is CN = 0.56. In our experiments, the presence of ~27.8% of Cu₄-O configuration species in OD-Cu NWs, which indicates an average coordination number of ~0.56 for Cu-O. This is consistent with our theoretical simulation results.

We have added corresponding discussions, highlighting the changes made. Additionally, **Fig. R3** have been included in the Supporting Information as **Supplementary Fig. 15**.

Fig. R3 | AIMD simulation of RDFs between Cu-Cu and Cu-O on B-OD-Cu(111) surface.

Alternatively, will the dynamically formed oxygen-containing species such as OH and surface oxygen contribute to EA formation activity?

Response: We appreciate the reviewer's insightful comment. On the one hand, dynamical OH adsorption during catalysis is a common phenomenon and both Cu NWs and OD-Cu NWs would adsorb OH or other O containing species in the electrolyte during catalytic process. However, since OH is dynamically adsorbed on both surfaces during catalytic process, the catalytic activities of Cu NWs and OD-Cu NWs are intrinsically different, suggesting the adsorption of OH is not the main factor to explain the performance difference.

On the other hand, DFT calculations were also performed to investigate the effect of dynamically adsorbed surface oxygen-containing species on the electrochemical reduction of acetonitrile to ethylamine. **Fig. R4 a-c** shows the configurations of Cu (111) and B-OD-Cu (111) surfaces with OH and O adsorption, and the adsorption energies of all intermediates involved in the electroreduction of acetonitrile to ethylamine. The results show that the adsorptions of amine intermediates on these surfaces with oxygen-containing species adsorption are slightly enhanced compared with pristine Cu (111) and B-OD-Cu (111) surfaces. The effect of surface-adsorbed oxygen-containing species on the reaction mechanism of the hydrogenation of acetonitrile to ethylamine is investigated. On Cu(111) surface, after introducing O* or OH*, the desorption of CH₃CH₂NH₂* remains as the potential-dependent step (PDS), which is similar to the pristine Cu(111). In comparison, on B-OD-Cu (111) surface, after adsorbing O* or OH*, the proton-coupled electron transfer electrochemical processes of CH₃CHNH* to CH₃CH₂NH* is still the PDS, consistent with B-OD-Cu (111) surface.

Meanwhile, the energy barriers on Cu(111) surface and its oxygen-containing surfaces ($O_{\text{ads}}@Cu(111)$ and $OH_{\text{ads}}@Cu(111)$) are always higher than on B-OD-Cu(111), $O_{\text{ads}}@B-OD-Cu(111)$ and $OH_{\text{ads}}@B-OD-Cu(111)$ surfaces. In a word, the dynamically formed oxygen-containing species would not change the PDS on either surfaces and would not change the trend of the energy barrier. Instead, the local surface electron distribution caused by lattice oxygen residues is believed to be the key factor to enhance the electroreduction of acetonitrile to ethylamine on the Cu metal surface.

We have added corresponding discussions, highlighting the changes made. Additionally, **Fig. R4** have been included in the Supporting Information as **Supplementary Fig. 41**.

Fig. R4 a, Configuration of Cu and OD-Cu catalysts adsorbing different oxygen-containing species ($OH_{\text{ads}}@Cu(111)$, $O_{\text{ads}}@Cu(111)$, $OH_{\text{ads}}@OD-Cu(111)$ and $O_{\text{ads}}@OD-Cu(111)$). **b,c** Adsorption energies of key intermediates involved in the acetonitrile reduction reaction on Cu(111) (**b**) and B-OD-Cu(111) (**c**) surfaces adsorbed by different oxygen-containing species. **d, e** Potential energy diagrams of the optimal formation route for the electroreduction of acetonitrile to ethylamine over Cu(111) (**d**) and B-OD-Cu(111) (**e**) catalysts adsorbed with different oxygen-containing species at 0 V.

4. *English writing needs improvement. The manuscript should be carefully proofread.*

Response: Thanks for your kind suggestion. We have double-checked the corresponding diagrams, tables and long sentences in the whole manuscript. In addition, authors have asked several colleagues to help improve the English.

REVIEWERS' COMMENTS

Reviewer #1 (Remarks to the Author):

This revised manuscript by Wang et al. discussed the residual lattice oxygen in OD-Cu NWS that plays a vital role in boosting the electroreduction of acetonitrile by facilitating the desorption of ethylamine. Previous comments have been well addressed and the authors have added new data and corrected several mistakes in the revised manuscript. I think the current version of manuscript is suitable to be published in Nature Communions.

Reviewer #2 (Remarks to the Author):

In this work, Wang and co-workers describe the promoted electrosynthesis of ethylamine from acetonitrile using OD-Cu NWs with abundant lattice oxygen. After carefully read the response, I think the authors have basically answered the reviewer's comments and questions, especially, the authors further clarify the novelty and difference compared to the published paper (Nat. Commun. 2021, 12, 1949, Chem Catal. 2021, 1, 393-406). They developed the high current especially at more than 1.0 A cm⁻², OD-Cu NWs exhibit an unprecedented Faradic efficiency (55.7 %), which displays the good practical potential. and they also clarified the different mechanism by in situ characterizations and theoretical calculations uncover, therefore, I think this manuscript further develop the conversion of acetonitrile. I agree this revised manuscript can be published.

Reviewer #3 (Remarks to the Author):

Although the authors devoted significant efforts to revise the manuscript by including additional characterizations to confirm the presence of "oxygen residuals", I am not convinced that the advancement is sufficiently significant to merit its publication in Nature Communication.

I am still puzzled by the authors' statements on page 18-19: "It is clear that due to the strong adsorption of CH₃CH₂NH₂*, the desorption of CH₃CH₂NH₂* is the potential-dependent step (PDS) of the reaction, requiring a high energy barrier of 0.69 eV." ... " the PDS is switched from the chemical process of CH₃CH₂NH₂* desorption into the proton-coupled electron transfer electrochemical processes".

First, "potential-dependent" should be "potential-determining". A nonelectrochemical desorption step may be rate-limiting but how can it ever be potential-determining?

Point-by-point response to the referees' comments

Reviewer #1 (Remarks to the Author): *This revised manuscript by Wang et al. discussed the residual lattice oxygen in OD-Cu NWS that plays a vital role in boosting the electroreduction of acetonitrile by facilitating the desorption of ethylamine. Previous comments have been well addressed and the authors have added new data and corrected several mistakes in the revised manuscript. I think the current version of manuscript is suitable to be published in Nature Communications.*

Response: We sincerely thank the referee for kindly suggesting acceptance for publication in *Nature Communications*.

Reviewer #2 (Remarks to the Author): *In this work, Wang and co-workers describe the promoted electrosynthesis of ethylamine from acetonitrile using OD-Cu NWs with abundant lattice oxygen. After carefully read the response, I think the authors have basically answered the reviewer's comments and questions, especially, the authors further clarify the novelty and difference compared to the published paper (Nat. Commun. 2021, 12, 1949, Chem Catal. 2021, 1, 393-406). They developed the high current especially at more than 1.0 A cm⁻², OD-Cu NWs exhibit an unprecedented Faradic efficiency (55.7 %), which displays the good practical potential. and they also clarified the different mechanism by in situ characterizations and theoretical calculations uncover, therefore, I think this manuscript further develop the conversion of acetonitrile. I agree this revised manuscript can be published.*

Response: Response: We sincerely thank the referee for kindly suggesting acceptance for publication in *Nature Communications*.

Reviewer #3 (Remarks to the Author): *Although the authors devoted significant efforts to revise the manuscript by including additional characterizations to confirm the presence of "oxygen residuals", I am not convinced that the advancement is sufficiently significant to merit its publication in Nature Communication.*

Response: We thank the reviewers for their positive feedback on the complete characterization to confirm "oxygen residuals" in OD-Cu NWs. As for the novelties and significances, we have outlined them below:

- (1) Our work uncovers that lattice oxygen in copper plays vital roles in modulating the key intermediate adsorption and boosting the catalytic conversion from nitriles to primary amines. By employing various advanced static and dynamic characterization techniques, we reveal that the high abundance of free electrons on the metallic copper leads to strong electronic coupling with amines and therefore hinders its catalytic kinetics, while electron localization effect created by lattice oxygen can restrict the flow of surface electrons in copper surface and further reduce the binding affinity to amines. In comparison with previous work, the developed strategy in this manuscript is conceptually new for nitrile reduction.
- (2) The developed OD-Cu NWs catalyst exhibits an unprecedented FE_{EA} of $\sim 97.8\%$ at -0.32 V (vs RHE). Meanwhile, the fabricated membrane electrode assembly (MEA) demonstrates the potential application of OD-Cu NWs for nitrile reduction to primary amines at industry-scale current ($\sim 80\%$ at 1.6 A), which represents the best performance among the ever-reported catalysts.
- (3) The developed nitrile hydrogenation strategy was also effective in manipulating the catalytic reduction of other nitriles including Cyclopropanecarbonitrile (CPN), 3-Hydroxypropionitrile (3-HPN), Butyronitrile (BN), and Pentanenitrile (PN), to primary amines, suggesting its generality toward the catalytic hydrogenation of nitriles.

Combining these merits, we thus reasonably believe the significances of our work is suitable for publication in *Nature Communications*.

1. I am still puzzled by the authors' statements on page 18-19: "It is clear that due to the strong adsorption of $CH_3CH_2NH_2^*$, the desorption of $CH_3CH_2NH_2^*$ is the potential-dependent step (PDS) of the reaction, requiring a high energy barrier of 0.69 eV." ... "the PDS is switched from the chemical process of $CH_3CH_2NH_2^*$ desorption into the proton-coupled electron transfer electrochemical processes". First, "potential-

dependent" should be "potential-determining". A nonelectrochemical desorption step may be rate-limiting but how can it ever be potential-determining?

Response: We really thank the referee for the careful review. We agree with the referee that "potential-determining" should be more precise. We have corrected it in the revised manuscript. In addition, to address the reviewer's question "*The non-electrochemical desorption step may be rate-limiting, but how can it be potential-determining?*", we have checked the concept of potential-determining step (PDS).¹ The PDS is the step with the highest energy barrier in the reaction thermodynamics, which is also considered as the "thermodynamic bottleneck"¹. Extra potential needs to be applied to overcome this thermodynamic barrier to drive this reaction. Thus, the non-electrochemical process with the highest energy barrier could be the PDS. For example, the non-electrochemical adsorption of O* and OH* in the oxygen reduction reaction is a PDS². In order to avoid confusion, we have revised the '*Statement on pages 18-19*' to make it clearer.

Reference

1. Koper, M. T. M. Analysis of electrocatalytic reaction schemes: distinction between rate-determining and potential-determining steps. *J. Solid State Electrochem.* **17**, 339-344 (2012).
2. Kulkarni, A., Siahrostami, S., Patel, A. & Norskov, J. K. Understanding Catalytic Activity Trends in the Oxygen Reduction Reaction. *Chem. Rev.* **118**, 2302-2312 (2018).